EMBO
Molecular Medicine

# Alpha-synuclein misfolding as fluid biomarker for Parkinson's disease measured with the iRS platform

Martin Schuler[1,2], Grischa Gerwert[1,2], Marvin Mann[1,2], Nathalie Woitzik[1,2], Lennart Langenhoff[1,2], Diana Hubert[1,2], Deniz Duman [ID][1,2], Adrian Höveler[1,2], Sandy Galkowski[1,2], Jonas Simon[1,2], Robin Denz[3], Sandrina Weber[4], Eun-Hae Kwon[5], Robin Wanka [ID][1,2], Carsten Kötting [ID][1,2], Jörn Güldenhaupt[1,2], Léon Beyer[1,2], Lars Tönges [ID][5], Brit Mollenhauer[4,6,7] & Klaus Gerwert [ID][1,2 ✉]

## Abstract

Misfolding and aggregation of alpha-synuclein (αSyn) play a key role in the pathophysiology of Parkinson's disease (PD). Despite considerable advances in diagnostics, an early and differential diagnosis of PD still represents a major challenge. We innovated the immuno-infrared sensor (iRS) platform for measuring αSyn misfolding. We analyzed cerebrospinal fluid (CSF) from two cohorts comprising PD cases, atypical Parkinsonian disorders, and disease controls. We obtained an AUC of 0.90 ($n = 134$, 95% CI 0.85–0.96) for separating PD/MSA from controls by determination of the αSyn misfolding by iRS. Using two thresholds divided individuals as unaffected/affected by misfolding with an intermediate area in between. Comparing the affected/unaffected cases, controls versus PD/MSA cases were classified with 97% sensitivity and 92% specificity. The spectral data revealed misfolding from an α-helical/random-coil αSyn in controls to β-sheet enriched αSyn in PD and MSA cases. Moreover, a first subgroup analysis implied the potential for patient stratification in clinically overlapping cases. The iRS, directly measuring all αSyn conformers, is complementary to the αSyn seed-amplification assays (SAAs), which however only amplify seeding competent conformers.

**Keywords** Parkinson's Disease Diagnosis; Alpha-synuclein in CSF; Protein Misfolding; Immuno-infrared-sensor; Differential Diagnosis
**Subject Categories** Biomarkers; Neuroscience

## Introduction

Parkinson's disease (PD) is a frequent neurodegenerative disorder that causes significant disability and an increasing global public health burden related to motor and non-motor features (Ben-Shlomo et al, 2024). It is characterized by a progressive loss of dopaminergic and non-dopaminergic neurons in the CNS (Feigin et al, 2018; Morris et al, 2024). Current PD diagnostic criteria mainly rely on the onset of clinical hallmark motor symptoms, including bradykinesia, rigidity, resting tremor, and postural instability. The disease is often advanced at the time of symptom onset, and over 50% of nigral dopaminergic neurons are already lost (Cheng et al, 2010). Unlike Alzheimer's disease (AD), blood-based biomarkers and positron emission tomography (PET) for diagnosis are subject to ongoing research and have not yet been well-established for αSyn (Smith et al, 2023; Korat et al, 2021). Therefore, objective biomarkers that accurately verify PD and distinguish it from other Parkinsonian disorders, ideally in the prodromal or earliest disease stages, are urgently needed and under discussion (Höglinger et al, 2024; Simuni et al, 2024). More specifically, the neuronal αSyn disease integrated staging system (NSD-ISS) based on biomarkers for neuronal αSyn, dopaminergic dysfunction, and genetic status, was proposed encompassing six stages (Simuni et al, 2024; Dam et al, 2024). While stage 0 is defined by the presence of fully penetrant pathogenic variants in genes (SNCA), stages 1 and 2 are preceding functional impairment and clinical diagnosis and are assessed with biomarkers for neuronal αSyn misfolding (1a) or dopaminergic dysfunction (1b) (Simuni et al, 2024; Dam et al, 2024). Stages 3–6 are characterized by functional impairment and clinical signs/symptoms, with increasing stages indicating increasing severity (Dam et al, 2024; Simuni et al, 2024).

The neuropathological hallmark of PD is the abnormal accumulation of αSyn in Lewy bodies (Shahmoradian et al, 2019). Various factors, such as genetic predisposition and post-translational modifications, are believed to contribute to the misfolding and aggregation of αSyn, resulting in the formation of oligomers, amyloid-like fibrils, and deposits such as Lewy bodies (Meade et al, 2019). In recent years, new body fluids techniques, including protein amplification assays such as Protein Misfolding Cyclic Amplification (PMCA) and the Real-Time Quaking-Induce Conversion (RTQuIC), now known under the consensus term of

[1]Center for Protein Diagnostics (PRODI), Ruhr-University Bochum, Bochum, Germany. [2]Department of Biophysics, Ruhr-University Bochum, Bochum, Germany. [3]Department of Medical Informatics, Biometry and Epidemiology, Ruhr-University Bochum, Bochum, Germany. [4]University Medical Center Göttingen, Department of Neurology, Göttingen, Germany. [5]Department of Neurology, St. Josef-Hospital, Ruhr-University Bochum, Bochum, Germany. [6]Paracelsus-Elena-Klinik, Klinikstraße 16, 34128 Kassel, Germany. [7]Scientific employee with an honorary contract at Deutsches Zentrum für Neurodegenerative Erkrankungen (DZNE), Göttingen, Germany. ✉E-mail: klaus.gerwert@ruhr-uni-bochum.de

seed amplification assays (SAA), have emerged (Concha-Marambio et al, 2021; Groveman et al, 2018; Shahnawaz et al, 2020). Previous studies have shown that these techniques detect aggregated and misfolded αSyn in cerebrospinal fluid (CSF) in clinical stages with accuracies over 90% (Rossi et al, 2020; Russo et al, 2021). However, these assays strongly rely on multiple factors (matrix, wells, buffers, etc.), including the quality of the reaction substrate for amplification, and comparability for quantification needs further harmonization, which is ongoing (Bellomo et al, 2022; Mammana et al, 2024).

Complementary, the immuno-infrared-sensor (iRS) platform measures the misfolding of the biomarker directly and label-free by use of infrared spectroscopy. This provides a direct measure of the secondary structure distribution of the respective fluid biomarkers by difference spectroscopy. Suppose the β-sheet misfolded conformers of β-amyloid in AD, are predominant compared to the α-helical/random-coil conformers. In that case, it indicates a high risk for a later AD clinical diagnosis in an early symptom-free stage (Nabers et al, 2016a). Misfolding of β-amyloid as a fluid biomarker for high AD risk is shown in individuals with AD, MCI, subjective cognitive decline, and symptom-free stages (Nabers et al, 2016b, 2018, 2019; Beyer et al, 2021; Stockmann et al, 2020). Importantly, past iRS studies for β-amyloid demonstrated robust CSF and blood matrix performance. In a real-world community-based cohort, high risk for clinical AD diagnosis was determined up to 17 years in advance in a symptom-free stage (Beyer et al, 2023). This study extends the iRS approach to αSyn misfolding in CSF in discovery and an independent validation study for identifying individuals with PD, atypical Parkinsonian syndromes, and disease controls.

## Results

### Controls

The iRS platform technology was applied to determine the secondary structure distribution of αSyn in CSF. The iRS platform and the capture antibody's performance were extensively characterized by the iRS platform itself and orthogonal techniques (ELISA, western blots, SPR) to qualify the analysis for CSF.

Figure 1A shows the binding of the different αSyn conformations by the capture antibody. In order to determine the secondary structure distribution of αSyn in body fluids, the catcher antibody must be able to bind all different conformers. The structure-sensitive Amide-I band showed its maximum at 1650 cm$^{-1}$ in the case of the α-helical/random-coil monomers. The maximum was at 1647 cm$^{-1}$ for dopamine-stabilized oligomers, while the most abundant oligomer species ranged from 700 to 1100 kDa (Stressmarc BioScience Inc., 2019). Pre-formed fibrils (PFFs) of αSyn showed a maximum at 1624 cm$^{-1}$. The results exhibit that the used catcher antibody extracts the different conformers. The expected range of αSyn extracted by capture antibodies from CSF is from 1624 to 1650 cm$^{-1}$. Common infrared spectroscopy wavenumber ranges for α-helices range from 1645 to 1662 cm$^{-1}$, for rando-coil motifs from 1640 to 1645 cm$^{-1}$, and for β-sheet motifs between 1615 and 1638 cm$^{-1}$ (Susi and Byler, 1986; Goormaghtigh et al, 1990).

The inertness of the functionalized ATR surface without the capture antibody is displayed in Fig. 1B. Excessive amounts of the different antigen conformers, at even much higher concentrations than the physiological concentrations observed in CSF, are blocked and do not bind unspecific to the surface. Notably, the surface has enough binding capacity to bind high amounts specifically by the antibody. Furthermore, cross-reactivity of abundant proteins like HSA or amyloidogenic proteins like β-amyloid (monomers and fibrils) do not bind even in excessive amounts on the antibody functionalized surface (Fig. EV3). Importantly, the functionalized blocking layer without capture antibody does not indicate an unspecific signal from the CSF matrix either (Fig. EV3). Taken together, the results demonstrate no unspecific binding or strong cross-reactivity from CSF to the functionalized surface.

Two independent control experiments were performed to determine the extraction of αSyn by the capture antibody. In the first experiment, an already measured CSF sample was measured a second time on a freshly prepared antibody functionalized surface. Figure 2A shows the kinetics (left) and spectra (right) for the first and second runs. No infrared signal is observed in the second run, implying complete extraction of αSyn from CSF on the surface within the infrared sensitivity in the first run. A minimal baseline shift (2. run) occurred in the time course of measurement (6 h) was observed but did not impede the interpretation of signals. A second experiment was performed to eliminate the possibility of remaining, but not measurable, amounts of αSyn in the iRS. A commercially available quantitative ELISA for total αSyn (BioLegend, Cat. No. 448607) was used. For this, the initial CSF and the CSF supernatants after measurement were compared either on a surface functionalized with or without capture antibody (Fig. 2B). Dilutions due to the iRS setup were adjusted. While the original CSF sample concentration was 1197 ± 37 pg/ml, the concentration in the supernatant decreased over 85% to 161 ± 14 pg/ml, implying a high efficiency in binding αSyn from CSF. Notably, the concentration of αSyn was only slightly decreased when measuring on a functionalized surface without capture antibody (1075 ± 19 pg/ml), which agrees with no signal being observed in the iRS read-out. The samples were measured in duplicates. The results show that the functionalized surface with the catcher antibody extracts almost all αSyn from the CSF.

In summary, the functionalized surface does bind specific αSyn and does not bind other CSF compounds unspecifically. Further antibody characterization (e.g., Appendix Fig. S1) displayed a favorable, balanced antibody binding profile with EC$_{50}$ values in the sub-nM range.

### Discovery and validation cohorts

The iRS platform technique was applied to $n = 134$ CSF samples from different clinical centers. The discovery study encompassed 59 individuals, with 17 out of 59 diagnosed with PD and 42 out of 59 as disease controls without signs of neurodegenerative disorders. The validation cohort comprised 75 individuals from the Paracelsus-Elena-Klinik (Kassel) and is entirely independent of the discovery cohort.

Patients with overlapping disorders were included in the validation cohort for a more challenging differential classification (details in Dataset EV1). In the Kassel cohort, 40 out of 75 were diagnosed with clinical PD. Since the presence of αSyn-aggregates and conclusively αSyn-misfolding in MSA patients is known, both PD and MSA subjects were considered as misfolding positive group, while all other subjects were grouped as one disease control

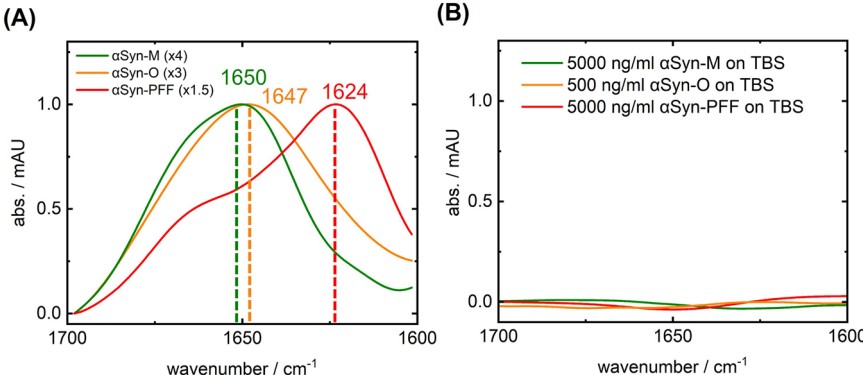

**Figure 1. iRS-surface characterization by synthetic αSyn antigens.**

(A) The secondary structure-sensitive Amide-I band absorbance of capture-antibody-bound αSyn-monomers (green, Stressmarq Bioscience Inc SPR-321), αSyn-oligomers (orange, Stressmarq Bioscience Inc SPR-466), and αSyn-PPFs (red, Stressmarq Bioscience Inc SPR-322) in PBS at high concentration of 500 ng/ml and scaled (x1.5–4) for comparison. Their structural differences are indicated by the significant wavenumber shift (cm⁻¹) ranging from 1650 cm⁻¹ for α-helical/random-coil (green) over 1647 cm⁻¹ (orange) to β-sheet dominated structures absorbing at 1624 cm⁻¹ (red). (B) The inertness measures on the blocking solution (BS) layer at high concentrations (500–5000 ng/ml) without the capture antibody. No signal is observed without the antibody on the blocking layer demonstrating sufficient inertness for pg-ng/ml concentrations of αSyn in CSF. αSyn alpha-synuclein, BS blocking solution, PFF pre-formed fibril. Source data are available online for this figure.

(control, compare Dataset EV1) group, including CBD, FTD, and PSP subjects (Schweighauser et al, 2020; Shahnawaz et al, 2020; Graves et al, 2023).

The diseased group showed an average downshifted maximum of $\upsilon_{av\_PD/MSA} = 1639.54\,\text{cm}^{-1}$ compared to the control group with a maximum of $\upsilon_{av\_control} = 1641.4\,\text{cm}^{-1}$. As read-out, not the absorbance maximum itself, but a center of mass maximum (upper 80–90% of the band) was taken, which increases robustness in small signals. A difference spectrum between all synucleinopathy (PD/MSA) and all control spectra averaged to one spectrum is performed to quantify the downshift. The normalized difference spectrum between the mean control spectra and the mean misfolding spectra marks a shift from random-coil/α-helical secondary structures at 1656.0 cm⁻¹ as a negative band to the β-sheet secondary structure at 1623.5 cm⁻¹ ratio as a positive band. The normalized and baseline-corrected difference spectrum is depicted in Fig. 3A. The positive/negative difference bands have comparable infrared signal integrals of 0.55 and 0.65. This shows clear-cut that the overall signal in the infrared reflects the expected transition for αSyn from random-coil/α-helical to β-sheet conformers. This misfolding is shown directly without amplification in body fluid for the first time as a native environment for αSyn in PD.

Changes in the region of 1700–1720 cm⁻¹ may indicate changes in unsaturated esters, ketones, or lipids that derive from metabolism changes (McMurry, 2023). Since the changes are visible on the iRS, they are derived from interaction partners of αSyn. The appearing band in PD/MSA individuals indicates the elevated presence of those species compared to controls.

In the next step, we used the 1656.0/1623.5 ratio from Fig. 3A as the best-performing measure instead of the downshift in wavenumbers to distinguish between controls and diseased. The ratio significantly distinguishes misfolding positive from negative cases ($P < 0.0001$; $P = 2.7 \cdot 10^{-16}$, Fig. 3B). Significance testing was done with a Mann–Whitney $U$ test, considering the non-conformity to a normal distribution (Appendix Fig. S4). A receiver-operating characteristic area under the curve (ROC-AUC) analysis with the

1656.0–1623.5 ratio is performed in the next step. In the discovery study, an area under the curve (AUC) of 0.90 (95% CI 0.85–0.96) is obtained, while in the validation study, an AUC of 0.86 (95% CI 0.80–0.93) is yielded (Fig. 3C). Combining both datasets yielded an AUC of 0.90 (95% CI 0.85–0.96, Fig. 3D). Notably, a logistic regression model without the iRS read-out and considering age and sex only yielded an AUC of 0.67 in this dataset (Appendix Fig. S2), clearly demonstrating the added value of the test.

The subgroup performance was analyzed (compare Fig. EV4) in the validation cohort. Because the subgroups were still very small (≤10), the data must be confirmed in future studies by larger numbers. The MSA patients alone could be differentiated from the control group with an AUC of 0.73 (95% CI 0.56–0.97) but worse from PD with an AUC of 0.71 (95% CI 0.49–0.93).

We introduced two thresholds instead of one, providing three classes (Fig. 4), as the misfolding negative and positive groups revealed an overlap. This reflects that the misfolding increases in a continuum between healthy controls and individuals with clinically confirmed synucleinopathies (PD/MSA), which is not accounted for by one threshold only. Individuals with a 1656.0/1623.5 ratio <1.065 were stated as a high misfolding (red) and clear synucleinopathy affliction, whereas a ratio of >1.14 reflected low misfolding (green) and no affliction. High- and low-misfolding groups depicted a significant difference ($P < 0.0001$, Fig. 4A). The individuals in the yellow group, between the two thresholds 1.065 and 1.14, fall into the intermediate area. These individuals are not as clearly assigned as individuals of high or low misfolding groups by iRS-readout. The intermediate group reflects the misfolding continuum between a healthy and diseased state. In this line, individuals categorized into the intermediate group may hold an elevated risk for developing synucleinopathies compared to unaffected individuals (green group) but are not yet clearly afflicted (red group). Comparing only individuals with high (red) and low (green) misfolding groups yielded an AUC of 0.95 (95% CI 0.89–1.01) with a sensitivity of 97% and specificity of 92% (Fig. 4B).

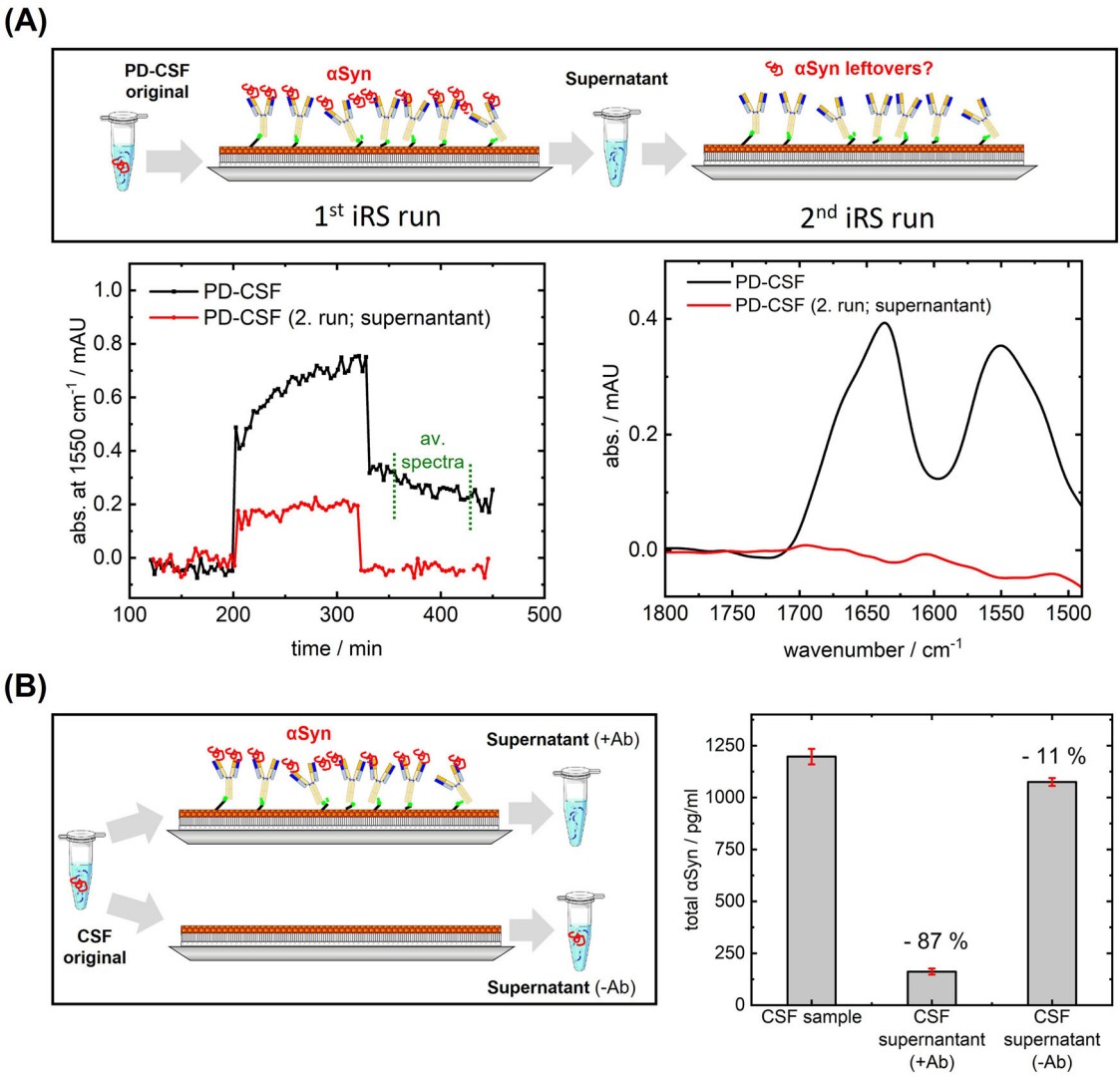

**Figure 2. iRS CSF sample kinetics and spectra of two subsequent runs of the same sample and indirect-ELISA experiments for quantification of αSyn reduction in samples with iRS-setup.**

(A) CSF sample signal as kinetic and averaged results spectra (averaging minutes 10–110 of wash) in two consecutive runs. The kinetic and sample signals of the first run show a signal, while the second run, regardless of the predilution, does not show any binding in the sample wash. (B) The setup of the ELISA experiment, where a CSF sample was circulated on a surface with and without the capture antibody. The original sample was withheld to be compared to the CSF supernatant collected after sample circulation. Dilution factors of iRS analysis were considered where needed. The original sample contained 1197 ± 37 pg/ml αSyn, while the supernatant of the CSF circulated in the presence of the capture antibody marks a decreased αSyn concentration of 161 ± 14 pg/ml. The αSyn concentration in supernatant without the capture antibody on the blocked surface is slightly reduced to 1075 ± 19 pg/ml. Samples and supernatants of the ATR ($n = 1$) were diluted according to the standard range and measured in ELISA duplicates. The standard deviation of duplicates is reported as error bars (red) with the mean values as the center (gray columns). Ab antibody, αSyn alpha-synuclein, CSF cerebrospinal fluid, iRS immuno-infrared-sensor. Source data are available online for this figure.

## Discussion

Disease-specific fluid biomarkers for PD and other neurodegenerative diseases are urgently needed to stratify affected patients. Atypical forms of the disease or associated co-pathologies with partially mimicking clinical syndromes may delay the clinical diagnosis and lead to suboptimal treatments. Robust biomarkers with high sensitivity and specificity that can be assessed *intra-vitam* are urgently needed since a confident diagnosis is currently only feasible through post-mortem neuropathologic examination. As another example, an αSyn-PET tracer will represent an important option and can serve as an important tool of proof for PD diagnosis, similar to the tracers already available for the detection of amyloid-β used in AD diagnostics. However, these tracers are still in early development (Smith et al, 2023; Korat et al, 2021).

In this study, we presented the disease-related misfolding of αSyn in CSF of PD/MSA patients as a promising fluid biomarker for the stratification of Parkinson's disease. Misfolding of αSyn has been previously linked to disease progression, neuronal cell death, and disease pathology (Lewy bodies and neurites) in numerous studies, and therefore αSyn demonstrates a promising biomarker candidate (Magalhães and Lashuel, 2022). Because misfolding of

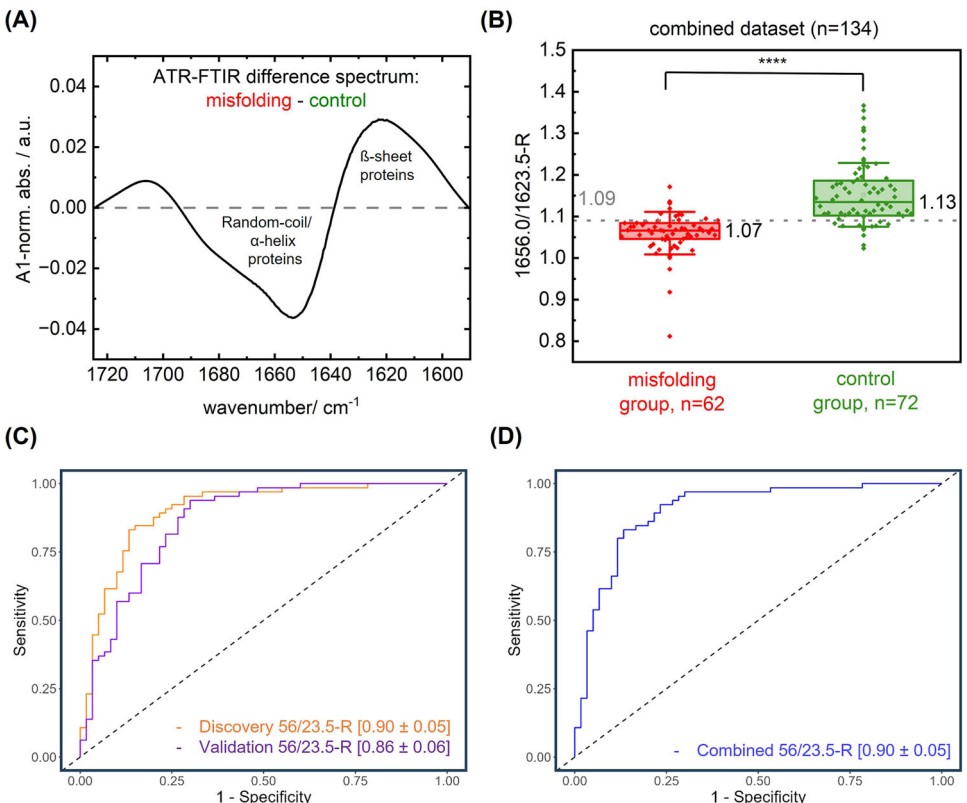

**Figure 3. iRS analysis of αSyn secondary structure distribution in CSF.**

(A) The group-level changes in the Amide-I-band as normalized and zoomed difference spectra. When calculating the difference spectra (normalized misfolding ($n = 62$) – control group ($n = 72$)), positive absorbance values at 1623.5 cm$^{-1}$ reflect increased β-sheet structures, while negative absorbance values at 1656.0 cm$^{-1}$ reveal the connected decrease in α-helical/random-coil structures for PD/MSA samples compared to control samples. (B) A Boxplot of the best-performing spectral feature (1656.0/1623.5 ratio) for the combined dataset, including 62 cases classified as misfolding positive (PD/MSA diagnosis) and 72 cases as controls where every point represents a single patient. A single threshold (1.093) discriminates both groups. A Mann–Witney *U* test revealed statistically significant group differences (****; CI = 95%, exact *P* value $2.7 \cdot 10^{-16}$). Box and whisker plots show median value (vertical line), interquartile range (boxes), and 1× standard deviation (whisker). The box minima and maxima are 1.01/1.11 (misfolding group) and 1.08/1.22 (control group). (C) The ROC-AUC analyses αSyn misfolding vs. disease controls (control) in the discovery (orange) and validation study (purple) utilizing the 1656.0/1623.5 cm$^{-1}$ spectral ratio and retrieved by a log. regression model considering age and sex. In the discovery study, an AUC of 0.90 ± 0.05 was reached, while the AUC in the validation study was at 0.86 ± 0.06. (D) The ROC curve of the combined dataset ($n = 134$), categorized into a misfolding group (PD/MSA) and a disease control group without expected misfolding. The ROC curve was retrieved by a log. regression model considering the 56/23.5 ratio, age and sex. The AUC value is 0.90 ± 0.05 (*P* value $5.6 \cdot 10^{-7}$). αSyn alpha-synuclein, CBD corticobasal degeneration, CSF cerebrospinal fluid, control, disease control, iRS immuno-infrared-sensor, PD Parkinson's disease, MSA Multiple System Atrophy, ROC-AUC receiver-operating characteristic area under curve. Source data are available online for this figure.

αSyn is expected already in NSD stage 1a, we propose that similar to misfolding of β-amyloid in AD, αSyn misfolding might also be a very early body fluid marker in prodromal or pre-motor PD for indicating at-risk status (Simuni et al, 2024; Concha-Marambio et al, 2023).

Promising results, indicating that misfolding of αSyn is an excellent fluid biomarker in Parkinson's disease, are obtained by SAA assays. They indirectly reflect the misfolding of αSyn by amplifying competent misfolded conformers with added monomeric αSyn in the SAA. This currently represents a promising PD test, mainly in CSF. The assay has reached high sensitivities and specificities (>90%) in CSF. However, the read-out depends on the recombinant monomeric αSyn, the competent amplified αSyn conformers, and the assay conditions (Bellomo et al, 2022). Therefore, a considerable variation in the read-out between different labs may be observed. A harmonization for future standardized applications is needed and ongoing because

quantitative read-outs of different tests developed should ideally be comparable, and expansions into peripheral matrices beyond CSF are examined (Bellomo et al, 2022; Mammana et al, 2024). Other quantification strategies for disease classification using reduction of αSyn concentration due to deposition in Lewy bodies could not discriminate between patients afflicted with the disease and controls (Magalhães and Lashuel, 2022).

Furthermore, there is an ongoing conceptual debate about whether PD and the spectrum of Parkinsonian disorders should be primarily regarded and differentiated as clinical diagnoses or if biomarkers and genetics should receive a much more prominent role (Schalkamp et al, 2023; Höglinger et al, 2024; Simuni et al, 2024).

Utilizing the iRS-platform, the synucleinopathies PD and MSA could be distinguished from controls by misfolding of αSyn when comparing only individuals with high (red) and low (green) misfolding groups yield an AUC of 0.95 (95% CI 0.89–1.01) with a

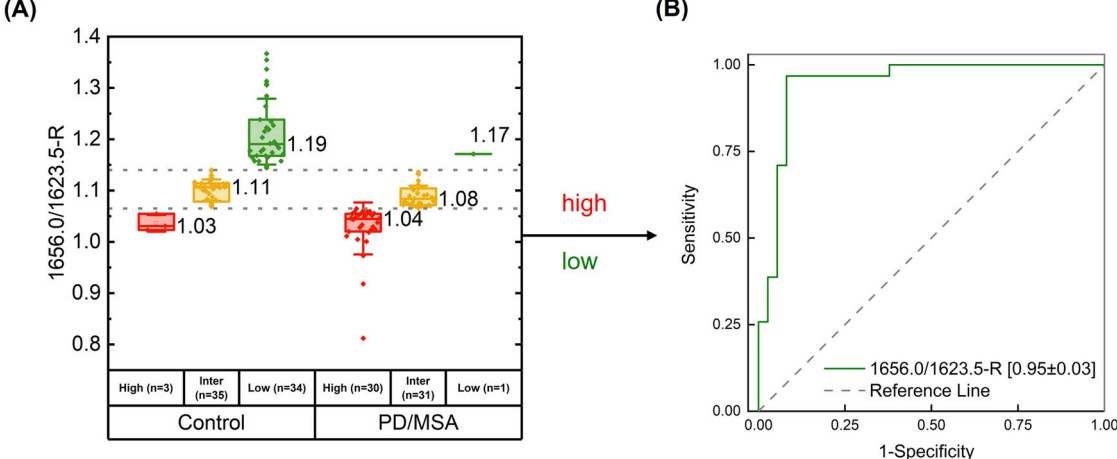

**Figure 4. iRS double threshold for classification into high (diseased) and low misfolding (controls) and an intermediate group at risk.**

(A) Classification for the control (*n* = 72) and PD/MSA *cohort* (*n* = 62). Within the control group and according to the selected thresholds (1.065/1.14), only 3 individuals fall into the high misfolding (false positive), 35 into the intermediate area (median 1.11), and 34 into the low misfolding control group. In contrast, in the PD/MSA group, 30 individuals fall into the high misfolding, 31 into the intermediate group (median 1.08), and only 1 (false negative) into the low misfolding group. Box and whisker plots include the median value (vertical line), interquartile range (boxes), and 1× standard deviation (whisker). The box minima and maxima are 1.02/1.05 (control, high), 1.08/1.12 (control, inter), 1.15/1.28 (control, low), 0.98/1.08 (PD/MSA, high), 1.07/1.11 (PD/MSA, inter), and 1.17/1.17 (PD/MSA, low). (B) The AUC performance of the high misfolding group (clearly affected) versus the low misfolding group (clearly not affected) (*n* = 68/134) according to the clinical diagnosis reached an AUC value of 0.95 ± 0.03 with a sensitivity of 97% and a specificity of 92%. In other words, two thresholds allow a clear distinction between Parkison/MSA and controls. The intermediate group is comprised of individuals at risk for conversion to the clinical stage and/or demonstrating overlap of αSyn misfolding pathology. AUC area under the curve, αSyn alpha-synuclein, iRS immuno-infrared-sensor. Source data are available online for this figure.

sensitivity of 97% and specificity of 92% (Fig. 4B) or by single decisive threshold (AUC 0.90, Fig. 3), which is similar to the accuracy from SAA. In contrast, an AUC of 0.67 is reached without the iRS read-out and only based on demographics (age, sex). This may be explained by the unbalanced gender distribution in the case of PD/MSA cases (M:F 77:23, compare Table 1).

Although high performance is achieved using the iRS misfolding analysis, a considerable group overlap of controls and PD/MSA can be concluded from Fig. 4A. This group may represent individuals at high risk or in prodromal stages, as preliminary iRS data of RBD patients was grouped in the intermediate group. Alternatively, the overlap may be caused co-pathology burden in controls. To address the overlap and challenge of a differential diagnosis due to the presence of well-known co-pathologies, a combination of biomarkers or additional misfolding analysis of key proteins like tau in atypical PD syndromes will improve biological stratification (Bellomo et al, 2022; Collij et al, 2024).

An in-depth analysis of the iRS spectra directly marks the transition from α-helical/random-coil to β-sheet in CSF (Fig. 3B). The secondary structure distribution of all different conformers of αSyn is measured with the iRS platform, whereas, in the SAA conformers, only the seeding competent conformers are amplified and provide the read-out. Oligomers with high α-helical content (compare Fig. 1) or β-sheet content of not fibrillar extended type may be underrepresented or not detectable in the SAA assay. The SAA platform only provides a binary positive or negative result, whereas the iRS result offers a measure of the continuum of disease progression. Although this needs to be investigated in further studies, this benefit could potentially answer the question of whether or not patients with prodromal PD (stage 1 NSD), e.g., REM sleep behavior disorder, progress to PD or associated αSyn

aggregation disease through an early risk indication in a disease continuum. So far, the inability to predict the timespan in which these patients develop PD is a significant hurdle for developing disease-modifying therapies and prevention studies (Mahlknecht et al, 2022).

Furthermore, the stratification of PD and the consideration of potentially overlapping neurodegenerative diseases in view of a cross-disease spectrum is essential for predicting a positive therapy response and facilitating the development of individualized therapies, as already seen for cancer. Therefore, secondary structure distribution profiles of overlapping syndromes may differentiate between neurodegenerative diseases, but these will require extended subgroup studies.

Secondary structure differences exist, e.g., in PD vs MSA-fibrils as resolved by cryo-EM. However, cryo-EM conditions in a vacuum at 77 K are rather artificial and might not reflect the secondary structure under physiological conditions and temperature. We believe that the iRS technology must confirm these findings under physiological conditions.

Concerning differential diagnosis of clinically overlapping syndromes, the subcohort of MSA patients was too small for reliable statements. It was reduced even more by two cases of an initial clinical diagnosis of MSA, which were changed to NPH and PSP but showed misfolding by iRS and additional SAA-positivity (Fig. EV4).

Literature reports of MSA cases with glial cytoplasmic inclusions (GCIs) as hallmarks have reported distinct seeding characteristics and altered β-sheet content if compared to PD pathology (Yamasaki et al, 2019; Araki et al, 2020; Okuzumi et al, 2023). SAA data of MSA cases suggested a shorter lag phase, higher seeding efficiency, and faster aggregation rate (Shahnawaz et al, 2020). Cryo-EM structures demonstrated distinct strains for MSA before (Schweighauser et al,

**Table 1. Study cohorts and sample demographics.**

| Parameter | PD/MSA | | | control | | | Statistics PD/MSA vs. control | | |
|---|---|---|---|---|---|---|---|---|---|
| | Dis. | Val. | Comb. | Dis. | Val. | Comb. | Dis. | Val. | Comb. |
| N total | 17<br>17 = PD | 45<br>40 = PD<br>5 = MSA | 62 | 42 = control | 30<br>5 = CBD<br>7 = FTD<br>9 = PSP<br>9 = control | 72 | – | – | – |
| Age [median ± SD] | 69 | 70 | 70 | 64 | 75 | 70 | MWU ($P = 0.243$) | MWU ($P = 0.181$) | MWU ($P = 0.540$) |
| Ratio of female [%] | 18 | 25 | 23 | 58 | 48 | 53 | Chi$^2$ $P$ <0.0001 | Chi$^2$ $P$ 0.0007 | Chi$^2$ $P$ 0.0001 |
| H&Y Scale % [mean ± SD] | 100 [2.6 ± 0.9] | 35 [3.2 ± 0.9] | 52 [2.9 ± 0.9] | 5 [3.5 ± 0.5] | 11 [2.3 ± 0.5] | 7 [2.8 ± 0.75] | – | – | – |
| UPDRS-III % [mean ± SD] | 23 [26 ± 10] | 94 [32 ± 10] | 74 [32 ± 10] | – | 48 [41 ± 14] | 19 [41 ± 14] | – | – | – |
| Q$_{Alb}$ pos.-rate % [mean + SD] (*) | 53 [9.5 ± 5.3] data 17/17 | 57 [10.4 ± 5.6] data 42/48 | 56 [10.1 ± 5.5] data 59/65 | 27 [6.7 ± 3.0] data 15/42 | 19 [7.7 ± 3.9] data 19/27 | 21 [7.3 ± 3.5] data 34/69 | – | – | – |

BBB blood–brain barrier, CBD corticobasal degeneration, control, disease control, FTD frontotemporal dementia, H&Y Hoehn & Yahr, MSA multiple system atrophy, MWU Mann–Whitney U, N number of participants, PD Parkinson's disease, UPDRS Unified Parkinson's Disease Rating Scale.
(*) Age-dependent albumin CSF/Serum quotient ($Q_{Alb} = Alb_{CSF}/Alb_{Serum}$) for assessment of BBB-dysfunction. Positivity for $Q_{Alb}$ was assumed when CSF/serum albumin quotient was higher than the calculated norm-value according to equation $Alb_{norm} = (4 + (age/15))$ (Farrall and Wardlaw, 2009; Albumin-Quotient (Liquor/Serum)).

2020). Interestingly, it was found that the content of β-sheet structures in Lewy Bodies in patients with PD was higher than in GCIs in patients with MSA (52.6 ± 1.9% vs. 38.1 ± 0.9%) by Fourier-transform infrared micro-spectroscopy (FTIRM) (Araki et al, 2020). In the case of the limited presented cases ($n = 5$), the ratio of the MSA subjects is closer to controls and not below PD cases (compare Fig. EV4), implying less β-sheets or other conformers in these cases. These results may align with the FTIRM literature, but be in contrast to other reports of the still ambiguous literature. Further direct comparisons of SAA and iRS with the same samples are strived to analyze if increased SAA activity correlates with lower iRS ratios and vice versa or if the readout possibly relies on a different set of species and therefore is not correlating.

In total, 57% (24/42) of the individuals diagnosed with PD or MSA in the validation cohort showed a BBB dysfunction, indicated by an albumin quotient (CSF/Serum) larger than the age-dependent norm value. 19 out of 24 (79%) showed an iRS-misfolding positive read-out. From the 18 negative cases, three were negative with the iRS, while 15 were positive for αSyn misfolding. In contrast, only 3 out of 27 controls had a BBB-dysfunction indicated; however, all were negative by iRS read-out. The correlation of disease diagnosis or misfolding-status positivity with BBB dysfunction may indirectly indicate a higher oligomeric/protofibril burden (Al-Bachari et al, 2020). Interestingly, 5 out of 24 of the patients with PD/MSA diagnosis and BBB-dysfunction present a negative misfolding read-out, which may be due to oligomers with less β-sheet content (compare Fig. 1) rather than the presence of β-sheet enriched protofibrils.

The iRS platform is well-stratified with orthogonal techniques. The platform extracts >85% of all αSyn out of CSF by the catcher antibody while unspecific binding is prevented (Figs. 2 and EV3). Only 10% of the analyte was lost without a capture antibody (Fig. 2). Since no iRS-signal was observed on the blocked layer, the loss may be due to the surrounding tubing or caused by not matching the dilution factor exactly, but not due to the functionalized surface itself.

All experiments are performed on home-built instruments, although precision, reproducibility, and device comparability are high (Fig. EV5; Appendix Fig. S3). In the future, the CE-certified commercial iRS1.1 instrument (betaSENSE) will improve these parameters further.

In addition to the observed changes of target proteins, natural binding partners of αSyn in CSF may contribute to the iRS read-out, possibly enhancing it. This would be the case if binding partners of misfolded and native fold αSyn are also different. A detailed analysis of the captured proteins will be performed to characterize the possible secondary binding partners using a newly developed and iRS-integrated mass spectrometry workflow.

We propose a combined iRS analysis of β-amyloid, tau, αSyn, and other biomarkers to address obstacles of co-pathologies and patient stratification based on protein misfolding (Nabers et al, 2019). Neuropathologic studies show that the incidence of mixed co-pathology is high among neurodegenerative disorders and modifies disease symptoms and progression (Robinson et al, 2018). Therefore, identifying co-pathology patterns could help better understand overlapping clinical symptoms and develop targeted therapy strategies.

Since misfolding is believed to occur before the onset of clinical symptoms, early diagnosis and risk prediction using the platform are addressed in a follow-up study, including isolated RBD patients (video-supported polysomnography verified) showing an enhanced risk for conversion to synucleinopathies (Nomura et al, 2012). For risk assessment and early stages, the double threshold classification has shown an intriguing result in the first subset of isolated RBD patients (Paracelsus-Elena-Klinik Kassel, Germany) and will be followed up.

To sum up, we developed the extension of the iRS technology from Alzheimer's to synucleinopathies, which for the first time allowed differentiation of synucleinopathies and controls by direct measurement of αSyn misfolding as conversion from α-helical to β-sheet by difference spectra in the unmodified CSF matrix. Further studies are in

progress concerning the biological classification (SynNeurGe frame network) of overlapping syndromes or co-pathologies in larger cohorts. Lastly, as misfolding is an early biomarker, iRS analysis of high-risk individuals (e.g., RBD) over time strives for risk assessment and prediction value.

# Methods

### Reagents and tools table

| Reagent/resource | Reference or source | Identifier or catalog number |
|---|---|---|
| **Antibodies** | | |
| Capture antibody 01: mouse IgG, monoclonal, epitope aa 121–140 (iRS analysis) | AC Immune | Not applicable |
| Anti-mouse IgG-HRP (Indirect ELISA EC$_{50}$) | Merck KGaA | Cat#A3673 |
| **Chemicals, enzymes, and other reagents** | | |
| Substrate (Indirect Elisa EC$_{50}$), Sigmafast OPD | Sigma-Aldrich Co. LLC | Cat#P9187 |
| 96-well plate (Indirect Elisa EC$_{50}$) | Greiner Bio-One GmbH | Cat#655076 |
| Microplate Shaker with Lid | VWR International LLC | |
| Kit for quantification of αSyn in CSF: LEGEND MAX™ Human α-synuclein | BioLegend | Cat#448607 |
| αSyn-Monomer, αSyn-Oligomer and αSyn-PFF antigens | Stressmarq | Cat#321, Cat#466, Cat#322 |
| Dulbecco's PBS (antigen preparation) | Invitrogen | Cat#14190136 |
| Surface functionalization buffers and solvents | WO2024003213A1 and WO2024003214A3 | Not applicable |
| **Software** | | |
| MATLAB, R2019b/R2024b | Mathworks | Not applicable |
| OriginPro 2021b/2024 | OriginLabs Corporation | Not applicable |
| OPUS 7.2 | Bruker | Not applicable |
| **Other** | | |
| Vertex 80 V | Bruker Optics Setup details: Refs 19–21 | Not applicable |
| ATR unit | Specac Ltd | Not applicable |
| CLARIOstar Plus | BMG LABTECH GmbH | Not applicable |

## Antigen preparations

Generation of αSyn pre-formed-fibrils (PFFs) was conducted according to the protocol of Polinski et al (Polinski et al, 2019). Briefly, αSyn stocks (Stressmarq, Cat. No. SPR-321), stored at −80 °C, were slowly thawed on ice. The protein solution was centrifuged at 14,400×*g* for 10 min at 4 °C to remove aggregates. The protein concentration of the supernatant was determined using the Bradford assay. Monomeric reference material aliquots were prepared in 1× Dulbecco's PBS (Invitrogen 14190136) with a concentration of 6.9 µM or concentrated for ThT-measurement (346 µM), snap-frozen in liquid N$_2$, and stored at −80 °C. Aliquots were only used once and diluted in PBS (pH 7.4) immediately before use and to the desired concentration.

For the generation of PFFs, peptide films were prepared to 346 µM in PBS, mixed, and incubated for 7 days at 37 °C under constant shaking at 1000 rpm. Successful fibrilization was reviewed by thioflavin T assay (compare Fig. EV1) and by secondary-structure-sensitive infrared spectroscopy against the buffer background. PFFs aliquots were snap-frozen in liquid N$_2$ and long-term stored at −80 °C or ambient temperature for use within one week. Next to the inhouse-produced fibrils, PFFs type I (Cat. No. SPR-322) from Stressmarq were acquired as reference material for comparison. In addition, human recombinant αSyn-oligomers (dopamine-HCl stabilized; Stressmarq. Cat. No. SPR-466) were aliquoted and stored at −80 °C upon arrival.

## Thioflavin T fluorescence for antigen verification

Antigen testing was performed for every generated or purchased batch. The indication of β-sheet structures in fibrillary (or some oligomeric) antigens and the absence of those in monomeric stock material was tested by a Thioflavin T (ThT) fluorescence assay. Antigen stock material, stored at −80 °C in single-use 346 µM aliquots, was thawed slowly on ice for triplicate ThT measurement.

In total, 25 µM ThT-PBS-solution (PBS: 137 mM NaCl, 7.8 mM Na$_2$HPO$_4$, 1.4 mM NaH$_2$PO$_4$, 3.2 mM KCl pH 7.4) was used for dilution of the antigen material (compare section "antigen preparation") to a final concentration of 8.6 µM in any wells of the 96-well-plate (Greiner Bio-One GmbH, Frickenhausen, Germany, Greiner 96 F-Bottom fluorescence plate, cat. no. 655076) assigned for measurement. Controls were included with sample buffer instead of sample (blank), monomeric material as negative control in case oligomeric or fibrillary material testing and vice versa. The plate was incubated for 60 min at room temperature while shaking at 300 rpm (VWR International LLC, Radnor, USA, Incubating Microplate Shaker with Lid). The read-out was performed by recording a fluorescence spectrum in the range of 410–460 nm (excitation scan, emission wavelength 490 nm) and 480–620 nm (emission scan, excitation wavelength 450 nm) using the plate reader (BMG LABTECH GmbH, Ortenberg, Germany, CLARIOstar Plus). Scan parameters included 20 flashes/well, a bandwidth of 10 nm, 300 rpm double orbital shaking for 5 s before plate reading, and a settling time of 0.5 s. The gain was adjusted at expected peak wavelengths (450/483 nm), and the focus was set to autofocus.

The presence of β-sheet rich aggregates was confirmed if the excitation scan maximum was shifted from 413 nm (free ThT) to 450 nm (intercalated ThT) and an emission increase by at least a factor of 20 was present when calculating the mean intensities at 482 nm against the blank without a sample (Polinski et al, 2019; Naiki et al, 1989). A representative example of the ThT spectra and calculation of the ThT-emission factor is presented in Fig. EV1.

## ATR-FTIR antigen spectra in buffer

For ThT-independent verification of the secondary structure features and estimation of α-helical content in β-sheet enriched antigens, the Bruker Alpha instrument (Bruker optics, part of Bruker Corp., Billerica, Massachusetts, USA) with a single reflection diamond ATR unit was used. In contrast to the measurements on specifically functionalized silicon crystals with capture antibody, the antigens on the Bruker Alpha were directly measured in buffer against the sample buffer as background. Typically, 10 background spectra (each 30 scans) were averaged and used recalculation of the averaged sample signal (10 spectra, each 30 scans). Spectra were recorded from 4000 to 400 $cm^{-1}$ and a spectral resolution of 2 $cm^{-1}$. Spectra were recorded using the OPUS software and then analyzed using the in-house MATLAB script to extract spectral features. The absolute Amide-I band maximum and spectral ratios were used to estimate β-sheet enriched structures compared to α-helical/random-coil monomers (compare Fig. EV1).

## Study cohorts

CSF samples of the discovery study were collected following standard diagnostic, clinical GCP, and current research guidelines and processed as previously described (Tashjian et al, 2019; Willemse and Teunissen, 2015; Mollenhauer et al, 2011). The experiments conformed to the principles set out in the WMA Declaration of Helsinki and the Department of Health and Human Services Belmont Report. Briefly, they were collected in the morning under fasting conditions, centrifuged, aliquoted, and stored at −80 °C on clinical sites. For the iRS analysis, they were provided in single-use 300 µl aliquots and measured in a blinded manner. Samples were transferred under dry-ice conditions to the site of iRS analysis and stored at −80 °C until measurement. CSF samples of the validation study were collected from one German PD specialty center using SOPs as published before (Paracelsus-Elena-Klinik Kassel, Germany) (Mollenhauer et al, 2017). Samples did not have artificial blood admixtures. The clinical diagnosis of Parkinson's disease (PD), multiple system atrophy (MSA), progressive supranuclear palsy (PSP), corticobasal degeneration (CBD), and frontotemporal dementia (FTD) were made according to current consensus criteria (Höglinger et al, 2017; Postuma et al, 2015; Wenning et al, 2022; Armstrong et al, 2013; Bott et al, 2014; Palmqvist et al, 2023). All samples are listed in detail in Dataset EV1.

Study subjects gave their informed consent, and study approval was obtained by the local ethics committee and institutional review boards (Bochum cohort institutional review board (IRB) number: # 17-6119; Kassel cohort: IRB Vote from Landesärztekammer Hessen: FF89/2008 and FF38/2016). Further samples have been obtained from routine outpatient care in a fully anonymized procedure for which the responsible ethics committee (Ethik-Kommission der Ärztekammer Westfalen-Lippe und der Westfälischen Wilhelms-Universität Münster, Germany) has waived ethical approval (file number 2022-692-f-N).

Table 1 summarizes the study cohorts and demographics, including age, sex, diagnosis, and additional information (H&Y scale, UPDRS-III, Albumin-Ratio ($Q_{Alb}$)) upon availability. After data acquisition and analysis, results were reported to cooperation sites, and clinical diagnosis and additional information were provided for performance evaluation.

## Determination of αSyn-misfolding by the iRS platform

In this approach, the structure-sensitive amide-I band of αSyn is read out in the infrared spectral region. This band reflects the C = O stretching vibration of the αSyn peptide backbone. The frequency of this band is downshifted when αSyn misfolds because the structure changes from a mainly random-coil/α-helical to β-sheet enriched secondary structure (Susi and Byler, 1986; Goormaghtigh et al, 1990). To subtract the large water background absorbance masking the much smaller αSyn absorbance, the attenuated total reflection (ATR) technology is applied. In the ATR approach, the evanescent wave generated by incident infrared light invades only about 500 nm of body fluid. This thin layer absorbance allows the subtraction of the water background absorbance, which is comparably large to sample signals and reveals by difference spectroscopy the αSyn absorbance spectrum (compare Fig. EV2).

The ATR unit is integrated into a Fourier-Transform infrared spectroscopy (FTIR) setup for the infrared absorbance measurements. For the ATR-FTIR measurements, Vertex 80 V spectrometers were used (Bruker Optics, Ettlingen, Germany) in combination with a liquid nitrogen-cooled mercury cadmium telluride (MCT) detector and a middle infrared (MIR) source. Device setup and spectrometer parameters for spectra acquisition were previously described in detail (Nabers et al, 2016a, 2016b, 2018). The ATR unit (Specac Ltd., Slough, England) was fit into the sample compartment of the FTIR instrument and aligned to an incidence angle of 45° (Nabers et al, 2016b). Measurements were conducted under constant dry airflow to prevent the sample chamber's sharp atmospheric water vapor absorbance bands (Nabers et al, 2016a, 2016b, 2018). As described before, multichannel measurements were performed with a motorized stage (Nabers et al, 2018).

## Antibody production

The native antibody with the epitope aa 131–140 was produced in large batches in a CHO cell line at AC Immune, purified, tested for affinity towards aS-M and aS-PFFs in internal Elisa setups and SPR, and provided for measurements by dry-ice shipping. Antibody was stored in aliquots at −80 °C in PBS buffer until usage. All CSF study samples were measured with the same antibody batch.

## Antibody-labeling and attachment to surface

A DBCO-containing molecule was attached to the antibody according to published protocols for covalent attachment to the functionalized surface (Broadpharm 2022; Sachin et al, 2012). The labeled antibody was compared to the native antibody in an indirect ELISA for $EC_{50}$ determination and on the sensor surface for antigen capture for quality control (compare Appendix Fig. S1).

## Indirect ELISA for $EC_{50}$ determination of (labeled) capture antibody molecules

The following protocol of an indirect ELISA for $EC_{50}$ value determination was used to assess the functionality of the distinct antibody batches before and after labeling with DBCO next to ATR-FTIR measurements. Next to this, the binding of antigens to the antibody was characterized externally by SPR and Elisa (AC Immune).

For the ELISA, the 96-well Nunc-Plate (Thermo Fisher Scientific Inc., Waltham, USA, cat. no. 439454) was coated with 1.5 µg/ml αSyn-M or αSyn-PFF (Stressmarq Bioscience Inc., Victoria, Canada, cat. no. SPR-321 and SPR-322) in 50 µl coating buffer (0.1 M Na$_2$CO$_3$, 0.1 M NaHCO$_3$, pH 9.7) for 1 h at 300 rpm shaking (Thermo Fisher Scientific Inc., Waltham, US, Wellwash™ Versa). The plate was washed thrice (0.1 M Tris, 0.15 M NaCl, 0.05% Tween20, pH 7.4), and wells were blocked with 1% skim milk in the wash buffer for 30 min. After repeated washing, the antibody to test was applied by a serial dilution in PBS starting from 750 nM for 1 h at 300 rpm shaking. After repeated washing, wells were incubated with the secondary antibody (anti-mouse IgG-HRP, Merck KGaA, Darmstadt, Germany, art. no. A3673) for 1 h. Lastly, and after washing, the substrate (Sigma-Aldrich Co. LLC, St. Louis, Missouri, US, art. no. P9187, Sigmafast™ OPD tablet) was automatically injected by needle (BMG LABTECH GmbH, Ortenberg, Germany, CLARIOstar Plus).

Absorbances were measured at 450 nm and 620 nm (background subtraction). For EC$_{50}$-determination, raw absorbances were blank and corrected using MARS software (BMG LABTECH GmbH, Ortenberg, Germany). After that, background absorbance values at 620 nm were subtracted from blank corrected values at 450 nm. The difference signal was used to calculate the EC$_{50}$ values with a software-embedded 4-parameter logistic fit model. An example of signal curves and calculated EC$_{50}$-values is presented in Appendix Fig. S1.

## Functionalization of the ATR surfaces

The αSyn absorbance spectrum is revealed by difference spectroscopy between αSyn bound spectra and spectra taken before αSyn binding, eliminating the large background absorbance spectra of the surface layer and the water background.

The surface functionalization has already been described in detail and schematically represented in Fig. EV2 (Nabers et al, 2016a, 2016b). In this study, an improved functionalization as described in the patents WO2024003213A1 and WO2024003214A3 was used, which will be described in detail elsewhere. The improvements generally led to optimized stability and inertness of the ATR surface compared to the former publications (Nabers et al, 2016a, 2016b). The surface functionalization was monitored by recording the respective infrared spectra in each step. This allowed precise control over each reaction step. The binding of all αSyn from the body fluid to the antibody-coated surface was conducted in a circulation mode. In contrast, subsequent washing steps in a flowthrough mode were used to rinse all unbound components present in the sample by buffer.

The capture antibody used in this study was extensively tested for binding of different αSyn-conformers covering monomeric, oligomeric, and fibrillary structures (example in Appendix Fig. S1). ELISA and external SPR analysis provided affinity information. The antibody was developed at AC Immune SA, Switzerland. Functionalized surfaces without antibodies were used to assess the specificity for αSyn and the inertness of the functionalized surface towards excessive amounts of antigen or CSF (Figs. 1B and EV3). Due to ongoing patenting processes, more details about the antibody characteristics cannot be shared and will be published elsewhere.

Continuous spectra recording during αSyn binding allowed tracking of minute changes in aqueous background or temperature.

One of four independent measurement channels was used as a reference channel by measuring a buffer instead of a sample, allowing for background monitoring and, if necessary, respective correction. After system stabilization in the PBS buffer, sample background spectra were measured. CSF samples were circulated over the surface for 120 min without additives. After sample circulation, the system was changed to flowthrough mode for sample washing, and loosely bound materials were rinsed away with PBS buffer. Sample wash spectra were recorded and reflected the αSyn bound spectra. For efficacy testing of the setup, the supernatants after circulation mode were collected and snap-frozen for concentration determination of αSyn.

## Colorimetric αSyn quantification in CSF by LEGEND MAX™ ELISA

For quantification of αSyn in original samples and measurement supernatants, the colorimetric kit LEGEND MAX™ Human α-synuclein from BioLegend® (San Diego, California, US, cat. no. 448607) was utilized with a reported sensitivity of 1.80 ± 0.21 pg/ml and a standard range of 10.2–650 pg/ml. All steps were conducted as stated in the manufacturer's protocol. CSF original samples were diluted 1:10 to match the standard range. Meanwhile, supernatants with a predilution of 1:3 due to ATR-FTIR measurement were adjusted to fit the original sample dilution before plate application. Samples and calibration curves were run in duplicates. Washing of the plate was programmed with the plate washer Wellwash™ Versa using 4× washing with each 300 µl/well. Absorbances at 450 and 570 nm (background) were read using the CLARIOstar Plus (BMG LABTECH GmbH, Ortenberg, Germany).

Plate reader settings included 22 flashes/well/cycle and two excitations at 450 and 570 nm measured after adding the stop solution. Overall, 300 rpm double orbital shaking for 5 s was done before read-out. The data was analyzed in the MARS software (BMG LABTECH GmbH, Ortenberg, Germany) by blank correction, background subtraction (450–570 nm), and a 4-parameter logistic fit model, as described before. The standard curves were used to calculate the absolute αSyn concentrations in original samples and supernatants after measurement on a capture-antibody-coated surface (= efficiency of setup) and on a surface without capture antibody (= system loss), as demonstrated in Fig. 2.

## Spectra processing and data analysis

Data analysis was performed with an in-house developed spectral software solution and scripts for MATLAB (Mathworks Inc., Natick, MA, USA, MatlabR2020b). For the detailed analysis, kinetics at specific wavenumbers indicating αSyn binding, e.g., at 1550 cm$^{-1}$ for bound protein, and whole spectra (4000–1480 cm$^{-1}$) indicating potential artifacts were analyzed. Raw spectra were averaged and corrected for water vapor and baseline. Difference spectra were calculated between the spectra before sample measurement (sample background) and the spectra of sample wash. The difference spectra were smoothened. The MATLAB script automatically generated spectral parameters, like the absolute amide-I-peak maximum, the amide-I center of mass maximum, or spectral ratios, e.g., 1656.0/1623.5 ratio. The different read-outs reflect the overall secondary structure distribution of αSyn. The analysis of the Amide-I/Amide-II-ratio at peak maxima is used as a quality indicator, and spectra were only included in a range of

### The paper explained

#### Problem

Parkinson's disease (PD) is the second most common neurodegenerative disorder that causes disability and public health burden. PD is a heterogeneous disease with overlapping syndromes and potential co-pathology overlap with other neurological disorders like Alzheimer's disease. Diagnosis is primarily based on clinical, symptomatic evaluation, including patient history, physical examination, and observation of motor symptoms such as tremors, rigidity, and bradykinesia. Especially, early stages of PD show subtle motor symptoms, and atypical features are often masked, making a clear diagnosis challenging. As a result, misdiagnosis or delayed diagnosis is common. There is an unmet medical need for fluid biomarkers to improve diagnostic accuracy based on a biological classification, as already established for Alzheimer's and starting in the PD field.

#### Results

This work presents a fluid biomarker that differentiates PD from non-motor disorder control individuals. In addition, it stratifies between idiopathic PD/multiple system atrophy (MSA) and atypical PD. The fluid biomarker is based on early-stage alpha-synuclein misfolding towards the disease-specific pathological conformers. The immuno-infrared sensor (iRS) readout indicates the secondary structure distribution of native and misfolded alpha-synuclein, which may be usable for a precise biological classification even at very early stages.

During the disease progression, the distribution shifted more and more from α-helical/random-coil toward β-sheet conformers. This allows continuous monitoring of the disease progression. The iRS readout shows 97% sensitivity and 92% specificity in the classification PD/MSA versus controls. In a traffic light scheme, these green (controls) and red (PD) groups, which are separated overlap with a yellow group which seem to have a high risk for progression towards the clinical motor stage or copathological features.

#### Impact

The structure-based biomarker, which is already established for Alzheimer's, now enables a biological classification for PD for timely identification and intervention, potentially slowing the disease progression and improving patient outcomes. In the future it could be used to monitor disease progression, guide treatment decisions, and accelerate the development of new drugs. Ultimately, very early identification and classification will enable trials to prevent larger PD motor symptoms by very early intervention.

1.10–1.50. Distorted spectra with a Ratio <1.10 or >1.50 due to minor water background absorbance changes or temperature instabilities during the recording of the sample spectra were excluded from the analysis. Spectra with a S/N < 20 were also excluded from the analysis. S/N was calculated with S as the mean absorbance between 1560 and 1540 $cm^{-1}$ and N as the root-mean-square of the absorbance values between 1800–1900 $cm^{-1}$ of unsmoothed spectra.

### Statistical analysis

Descriptive statistics were used to summarize patient characteristics to compare the age and sex of αSyn misfolding cases and controls by chi-square and Mann–Witney *U* tests.

Receiver-operating characteristic (ROC) analyses, including calculation of area under the curve (AUC), were conducted for discrimination of individuals with αSyn misfolding and controls

and by use of the ratio of 1656 $cm^{-1}$ and 1623.5 $cm^{-1}$, representing the distribution of monomeric α-helical/random-coil and β-sheet enriched protein structures. First, a logistic regression model was calculated with the disease status as the dependent variable and age, sex, and the read-out value as independent variables. The resulting model was then used to predict the probability that each person had Parkinson's disease. The ROC Curves and associated AUC values were then calculated based on these predictions. We additionally performed the same analysis stratified by study (Discovery and Validation, see Fig. 3). Moreover, the added value of the read-out was analyzed by comparison to age and sex model only (Appendix Fig. S2).

All analyses were conducted two-sided at a significance level of 0.05 using OriginPro 2021b or OriginPro 2024 (OriginLab Corporation, Northampton, MA, USA) and the programming language R (version 4.2.1).

## Data availability

This study includes no data deposited in external repositories. The αSyn antibody used in the research is commercially available to colleagues in academic research upon request to betaSENSE.

The source data of this paper are collected in the following database record: biostudies:S-SCDT-10_1038-S44321-025-00229-z.

## Peer review information

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

## Acknowledgements

AC Immune SA developed and produced the αSyn antibody used in this study. The authors greatly thank AC Immune SA for its provision and support in all antibody-related topics. The authors thank betaSENSE GmbH for providing the antibody, antibody-related protocols, and staff. The presented research was funded by the Center for Protein Diagnostics (PRODI), Ministry of Culture and Science of North-Rhine Westphalia, Germany.

## Author contributions

**Martin Schuler**: Conceptualization; Data curation; Formal analysis; Validation; Investigation; Visualization; Methodology; Writing—original draft; Writing—review and editing. **Grischa Gerwert**: Data curation; Investigation; Methodology; Writing—original draft. **Marvin Mann**: Data curation; Investigation; Methodology; Writing—original draft. **Nathalie Woitzik**: Data curation; Investigation; Methodology; Writing—original draft. **Lennart Langenhoff**: Investigation; Methodology. **Diana Hubert**: Investigation; Methodology. **Deniz Duman**: Investigation; Methodology. **Adrian Höveler**: Investigation; Methodology. **Sandy Galkowski**: Investigation; Methodology. **Jonas Simon**: Investigation; Methodology. **Robin Denz**: Software; Formal analysis; Validation; Visualization; Writing—original draft. **Sandrina Weber**: Data curation; Formal analysis; Writing—original draft. **Eun-Hae Kwon**: Data curation; Formal analysis; Writing—original draft. **Robin Wanka**: Methodology; Writing—original draft. **Carsten Kötting**: Conceptualization; Supervision; Methodology; Writing—original draft; Writing—review and editing. **Jörn Güldenhaupt**: Conceptualization; Software; Formal analysis; Supervision; Methodology; Writing—original draft; Writing—review and editing. **Léon Beyer**: Conceptualization; Supervision; Investigation; Methodology; Writing—original draft; Project administration; Writing—review and editing. **Lars Tönges**: Conceptualization; Resources; Data curation; Supervision; Writing—original draft; Project administration; Writing—review and editing. **Brit Mollenhauer**: Resources; Data curation; Supervision; Writing—original draft; Project administration; Writing—review and editing. **Klaus Gerwert**: Conceptualization; Resources; Supervision; Funding acquisition; Writing—original draft; Project administration; Writing—review and editing.

Source data underlying figure panels in this paper may have individual authorship assigned. Where available, figure panel/source data authorship is listed in the following database record: biostudies:S-SCDT-10_1038-S44321-025-00229-z.

## Funding

## Disclosure and competing interests statement

K.G. is the founder and CEO of betaSENSE GmbH. The remaining authors declare no competing interests.

# Expanded View Figures

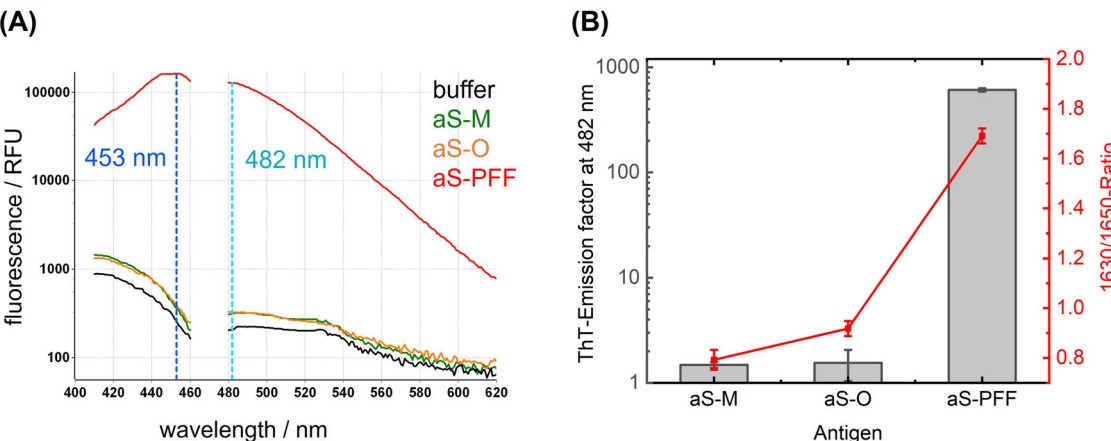

**Figure EV1.  ThT analysis of ∝Syn antigens.**

**A** The mean ThT-excitation and emission spectra from sample triplicates, recorded from 410–460 nm and 480–620 nm and displayed in relative fluorescence units (RFUs). For this, technical triplicates of αSyn-M (Stressmarq Bioscience Inc., SPR-321), αSyn-O (Stressmarq Bioscience Inc., SPR-466), and αSyn-PFFs (Stressmarq Bioscience Inc., SPR-322) were prepared with a final concentration of 7.14 µM in 25 µM ThT-PBS buffer, incubated for 60 min at ambient temperature in the 96 F-bottom fluorescence plate and measured with the plate reader in spectral scan mode. The excitation scan was performed from 410–460 nm, while the emission scan was performed from 480–620 nm, with a bandwidth of 10 nm and 20 flashes/well. The gain was adjusted at expected maxima (453/482 nm), and autofocus obtained focal height. Before read-out, the plate was shaken at 300 rpm for 5 s. (**B**) Mean ThT intensities at 482 nm were calculated from triplicate ThT measures of the respective antigens. The ThT-emission factor was calculated with mean intensities at 482 nm by division of mean intensities of the buffer at 482 nm (25 µM ThT-PBS). Moreover, the 1630/1650-R, derived from ATR-FTIR measurement in duplicates, was plotted against the ThT factor since the increased 1630/1650-R shows increased ß-sheet content of the given protein and decreased α-helical/random-coil content. To obtain the 1630/1650-R without the influence of binding tendencies of antibodies, antigens were measured directly on the Bruker alpha ATR-FTIR cell against the respective buffer (PBS, pH 7.4). The 1650/1630-R was retrieved automatically by the same MATLAB script used for the CSF samples after averaging 10 sample spectra for increased S/N. Error bars display the standard deviation of the ATR-duplicates or ThT-triplicates.

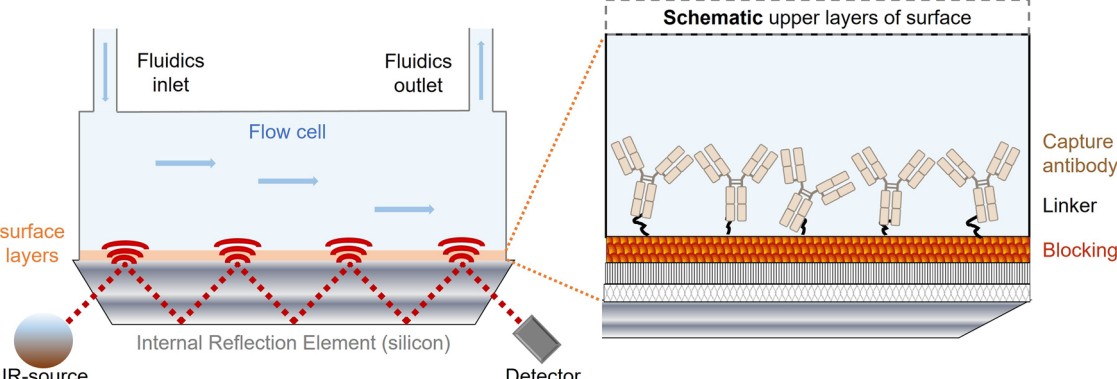

**Figure EV2.  Scheme of immuno-infrared sensor (iRS).**

The concept was published before in detail ((Nabers et al, 2016a, 2016c, 2018; Schartner et al, 2013)). The body fluid sample (without additives or modifications) is circulated in the sample chamber over the functionalized crystal (internal reflection element) surface in an ATR (Attenuated Total Reflection) setup ((A); image not to scale). At points of total reflection, an evanescent field (red) is generated invading only about 500 nm and preventing large water absorbance in the amide-I region (Goormaghtigh et al, 1990). The area of the evanescent field marks the measurable area in the iRS setup. (B) (image not to scale): The approach for surface functionalization is described in full detail (WO2024003213A1 and WO2024003214A3). The surface of the internal reflection element is functionalized with a capture antibody, which extracts all αSyn conformers from the body fluid. The antibody is bound by a linker to a blocking layer on the surface of the iRS. The blocking layer prevents unspecific binding on the surface of the iRS. The evanescent infrared beam measures the absorbance maximum of the amide-I band as read out. The αSyn absorbance is then elucidated by difference spectroscopy. For this, a background reference spectrum is taken before sample application and subtracted as background reference. The difference spectrum reveals the absorbance spectrum of all bound αSyn conformers without background from prior surface layers due to functionalization and water. Depending on the distributions of the conformers, the read-out varies between ~1655 cm$^{-1}$ wavenumbers for a pure α-helical/random-coil distribution and ~1625 cm$^{-1}$ for a pure β-sheet conformer distribution. Mixtures absorb in between depending on the conformer distribution. The downshift of the read-out represents the disease progression in a continuum by the secondary structure distribution of αSyn.

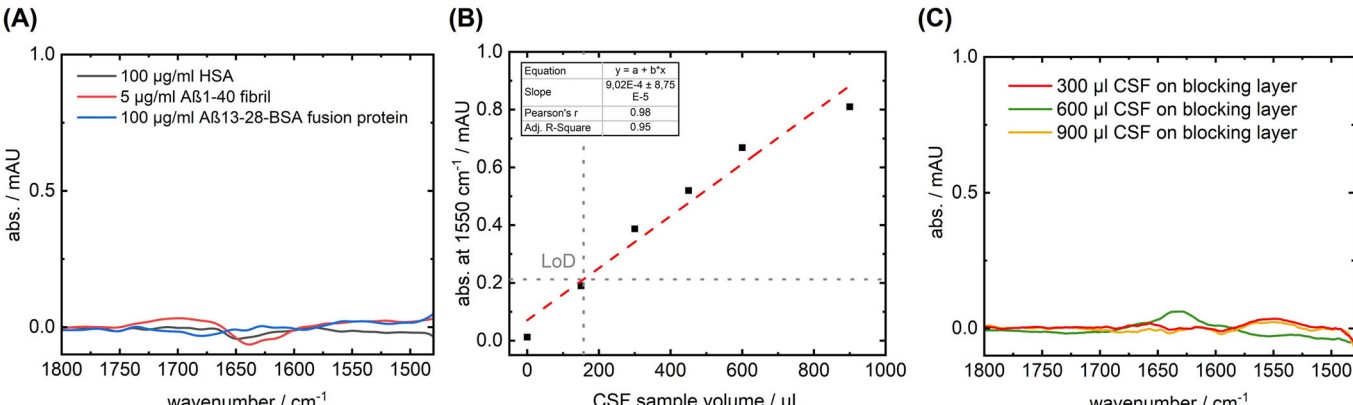

**Figure EV3. Testing of unspecific binding to the blocking layer.**

(A) The cross-reactivity test by mean wash spectra in PBS on an αSyn -capture antibody surface with HSA, Aß$_{1-40}$-fibrils, and Aß$_{13-28}$-BSA fusion protein (produced by two-step sulfo-NHS reaction with primary amines; Thermo Fisher Scientific™, art. no. 39269) in 5–100 µg range in 1 ml total volume. No bound signal is observed in all cases; thus, no cross-reactivity to abundant HSA and other amyloidogenic proteins such as β-amyloid in the form of a ß-amyloid$_{1-40}$ fibril (preparation according to Shin et al) or conjugated β-amyloid$_{13-28}$ (monomeric form) in high concentrations is observed (Shin et al, 1997). (B) The amount of immobilized protein on an αSyn capture antibody surface by absorbance at 1550 cm$^{-1}$ in relation to the used CSF volume. The assay signal between 300–900 µl shows alignment to linear fit, implying enough capture antibody molecules for the target analyte as expected for antibody excess surfaces. The limit of detection (LoD) was calculated by LoD = 3.3 (σ/s) with σ = standard deviation and s = slope / coefficient of X-variable to LoD = 3.3·(0.043/9.02·10$^{-4}$) = 157 µl CSF. Since S/N is critical when determining the secondary structure-sensitive amide-I spectral features and the aSyn amount in samples vary naturally, CSF volumes of 300 µl were analyzed for each sample. (C) The inertness of the blocking layer without the capture antibody on the surface for 300–900 µl CSF sample volume as mean wash spectra in PBS. No specific signals were observed at 1550 cm$^{-1}$ ( = bound protein) or in the amide-I region (1600–1700 cm$^{-1}$) without the capture antibody on the assay's blocking layer.

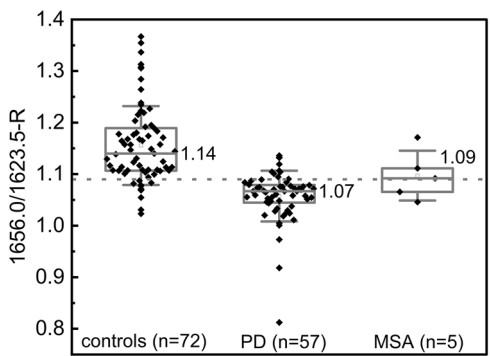

**Figure EV4.  Subgroup comparison of PD and MSA samples to controls.**

The five presented MSA cases are closer to controls than PD subjects regarding the iRS-readout measuring the αSyn misfolding. The two synucleinopathies, MSA and PD, are overlapping, although the number of MSA cases is still not significant yet. Box and whisker plots show median value (vertical line), interquartile range (boxes), and 1x standard deviation (whisker), while the given values report the median value of the groups. The box minima and maxima are 1.08/1.23 (controls), 1.01/1.11 (PD), and 1.05/1.15 (MSA). Note: small and unbalanced datasets in the case of MSA hinder robust interpretation of group differences.

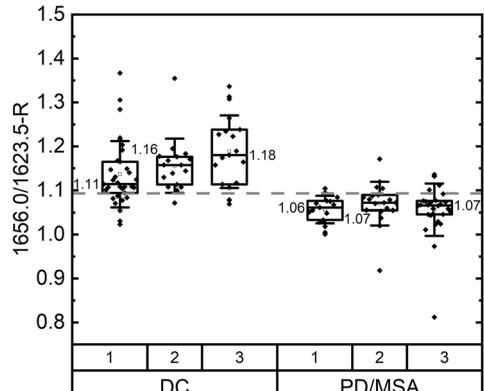

| M.-W. U: exact p-values | PM1 (n=17) | PM2 (n=19) | PM3 (n=29) |
|---|---|---|---|
| DC1 (n=33) | $1.5 \cdot 10^{-6}$ (****) | $1.8 \cdot 10^{-4}$ (***) | $9.9 \cdot 10^{-7}$ (****) |
| DC2 (n=17) | $3.9 \cdot 10^{-8}$ (****) | $4.1 \cdot 10^{-6}$ (****) | $3.2 \cdot 10^{-8}$ (****) |
| DC3 (n=19) | $1.6 \cdot 10^{-7}$ (****) | $9.1 \cdot 10^{-6}$ (****) | $9.1 \cdot 10^{-9}$ (****) |

**Figure EV5. Sample distribution ($n = 134$) on three independent iRS instruments (1,2,3).**

Different devices with randomized sample sets are comparable, and separation is highly significant. The whiskers of Boxplots depict 1× standard deviation, while the given values report the median value of the groups. The box minima and maxima are 1.06/1.21 (DC, 1), 1.09/1.22 (DC, 2), 1.11/1.27 (DC,3), 1.03/1.09 (PD/MSA, 1), 1.02/1.12 (PD/MSA, 2), and 1.00/1.12 (PD/MSA, 3) All subgroup comparisons by Mann–Whitney U testing are listed in the adjacent table (p values between $1.8 \cdot 10^{-4}$ to $9.1 \cdot 10^{-9}$).

