## [Peer Review File · EMBO Molecular Medicine]

Alpha-synuclein misfolding as fluid biomarker for Parkinson's disease measured with the iRS platform

Martin Schuler, Grischa Gerwert, Marvin Mann, Nathalie Woitzik, Lennart Langenhoff, Diana Hubert, Deniz Duman, Adrian Höveler, Sandy Galkowski, Jonas Simon, Robin Denz, Sandrina Weber, Eun-Hae Kwon, Robin Wanka, Carsten Kötting, Jörn Güldenhaupt, Léon Beyer, Lars Tönges, Brit Mollenhauer, and Klaus Gerwert

Corresponding author: Klaus Gerwert (klaus.gerwert@ruhr-uni-bochum.de)

Review Timeline:

Submission Date:	28th Oct 24
Editorial Decision:	4th Dec 24
Revision Received:	17th Jan 25
Editorial Decision:	4th Mar 25
Revision Received:	20th Mar 25
Accepted:	27th Mar 25

Editor: Jingyi Hou

Transaction Report:

4th Dec 2024

Dear Dr. Gerwert,

Thank you for submitting your work to EMBO Molecular Medicine. I would like to apologise for the slow process, which was due to the late arrival of reviewers' reports. We have now heard back from two of the three reviewers who agreed to evaluate your manuscript. Unfortunately, after a series of reminders we did not manage to obtain a report from Reviewer #1. In the interest of time, and since the recommendations of the other two reviewers are quite similar, I prefer to make a decision now rather than further delaying the process. If we receive the comments from Reviewer #1, we will send them to you, and you can address the issues raised by Reviewer #1 together with those raised by the other two reviewers. You will see from the comments below that both Reviewers #2 and #3 find the manuscript to be of interest. They raise, however, several important points, which should be convincingly addressed in a revision of this work.

I think that the recommendations of the reviewers are rather clear so there is no need to repeat the points listed below. All issues raised by the reviewers need to be satisfactorily addressed. As you may already know, our editorial policy allows in principle a single round of major revision, so it is essential to provide responses to the reviewers' comments that are as complete as possible. Please feel free to contact me in case you would like to discuss in further detail any of the issues raised by the reviewers.

Please also contact us as soon as possible if similar work is published elsewhere. If other work is published, we may not be able to extend the revision period beyond three months.

I look forward to receiving your revised manuscript soon.

Kind regards,
Jingyi

Jingyi Hou
Editor
EMBO Molecular Medicine

We require:

- 1) A .docx formatted version of the manuscript text (including legends for main figures, EV figures and tables). Please make sure that the changes are highlighted to be clearly visible.
- 2) Individual production quality figure files as .eps, .tif, .jpg (one file per figure). For guidance, download the 'Figure Guide PDF': (<https://www.embopress.org/page/journal/17574684/authorguide#figureformat>).
- 3) A .docx formatted letter INCLUDING the reviewers' reports and your detailed point-by-point responses to their comments. As part of the EMBO Press transparent editorial process, the point-by-point response is part of the Review Process File (RPF), which will be published alongside your paper.
- 4) A complete author checklist, which you can download from our author guidelines

(<https://www.embopress.org/page/journal/17574684/authorguide#submissionofrevisions>). Please insert information in the checklist that is also reflected in the manuscript. The completed author checklist will also be part of the RPF.

6) It is mandatory to include a 'Data Availability' section after the Materials and Methods. Before submitting your revision, primary datasets produced in this study need to be deposited in an appropriate public database, and the accession numbers and database listed under 'Data Availability'. Please remember to provide a reviewer password if the datasets are not yet public (see <https://www.embopress.org/page/journal/17574684/authorguide#dataavailability>).

.

- the medical issue you are addressing,

- the results obtained and

- their clinical impact.

12) Author contributions: You will be asked to provide CRediT (Contributor Role Taxonomy) terms in the submission system. These replace a narrative author contribution section in the manuscript.

13) A Conflict of Interest statement should be provided in the main text.

14) Every published paper now includes a 'Synopsis' to further enhance discoverability. Synopses are displayed on the journal webpage and are freely accessible to all readers. They include a short stand first (maximum of 300 characters, including space) as well as 2-5 one-sentences bullet points that summarizes the paper. Please write the bullet points to summarize the key NEW findings. They should be designed to be complementary to the abstract - i.e. not repeat the same text. We encourage inclusion of key acronyms and quantitative information (maximum of 30 words / bullet point). Please use the passive voice. Please attach these in a separate file or send them by email, we will incorporate them accordingly.

15) All Materials and Methods need to be described in the main text using our 'Structured Methods' format. According to this format, the Methods section includes a Reagents and Tools Table (listing key reagents, experimental models, software and relevant equipment and including their sources and relevant identifiers) followed by a Methods and Protocols section describing the methods, ideally using a step-by-step protocol format. The aim is to facilitate adoption of the methodologies across labs.

Please download and fill our Reagents and Tools Table template (.docx), which you can find in our author guidelines: <https://www.embopress.org/page/journal/17574684/authorguide#structuredmethods>

***** Reviewer's comments *****

Referee #2 (Remarks for Author):

The manuscript by Schuler et al. builds on the previous work by the Gerwert group in which they developed novel immunoinfrared-biosensors to detect the misfolding state of Abeta peptides in body fluids. The novelty in this paper is the capturing of alpha-synuclein by a specific monoclonal antibody and the detection of random-coil/alpha-helical or beta-sheet secondary structures based on infrared (IR) spectra shifts. After the technical description of the sensor development and initial tests, authors then set out for a proof-of-concept to explore whether the sensor could detect differences in IR spectra measured in cerebrospinal fluid (CSF) samples of patients with a synucleinopathy (ie, Parkinson's disease and multiple system atrophy, MSA) versus healthy individuals or those that are not diagnosed with a different neurological disorder, unrelated to a typical synucleinopathy. Using two sources of human samples, the authors report a change in the secondary structure distribution for alpha-synuclein in the group of individuals with a synucleinopathy which was different in the so called 'control group'. The 1656.0/1623.5 cm⁻¹ spectral ratio was then used together with age and sex for a log regression model to assess the strength to predict the presence of misfolded alpha-synuclein in a ROC-AUC analysis. Indeed the identified changes in the spectral ratio was suitable for a strong differentiation and prediction of people with a synucleinopathy in the tested samples. This is a first Proof-of-Concept that this biosensor has a certain value for a diagnostic test for synucleinopathies that now needs to be developed. The strength of this biosensor over other assays is that direct detection without modification of the sample is needed. This is in contrast to the currently being promoted Seed-Amplification-Assays (SAA). The IR based sensor also detects all forms of alpha-synuclein whereas the SAA requires the presence of a seed that amplifies recombinant alpha-synuclein substrate.

This manuscript is of broad interest to the Parkinson's research and clinical community and may also excite non-experts of that field because of the novel approach that is taken to measure a protein folding change in a complex biofluid. To better understand some aspects of the paper it would be good if the authors address following points:

1) The authors state that the alpha-synuclein antibody was provided by AC Immune. Normally, when a new antibody is published, more characterization is provided, such as the epitope, the binding affinity/KD, and binding kinetics by SPR. The authors mention on page 6 this information is available. Please provide this information. If it cannot be provided, please state clearly why not. The scientific community would profit from knowing more about this alpha-synuclein antibody and appreciates that AC Immune is making it available.

2) Also, conjugating antibodies often leads to changed binding properties. For a better understanding about availability and orientation of the antibody on the chip (cartoons show an C-terminal binding of the Fc part to the chip and the Fab part looks up, is that correct?), please indicate how many DBCO molecules are labeled per antibody molecule and also show data that the

DBCO-labeled antibody has the same binding properties as the unlabeled antibody.

3) Also, what is the density of the antibody on the chip? In the cartoon of Fig 2S it looks densely packed whereas the cartoon in Fig 2 shows a much less densely packed surface. It is suggested using the same cartoon for illustration to avoid confusion for the read.

At a certain density one would also expect more avidity effects and possible competition of binding aggregated alpha-synuclein with beta-sheets over monomeric alpha-helical alpha-synuclein. How is this considered for the 1656.0/1623.5 cm⁻¹ spectral ratio calculations? Please add to the discussion.

4) Figure 2A, right panel shows the effect of how alpha-synuclein was removed from the CSF sample by exposing it to the chip containing an alpha-synuclein antibody. The graph shows a prominent elevation of the absorption at 1550 cm⁻¹ and a disappearance of that spectra after removing of alpha-synuclein. Please add to the results this observation and explain what it means. Also, what happens at 1550 cm⁻¹ which was omitted on Figure 3A?

5) Figure 3A: At >1700 cm⁻¹ an elevation of the delta signal is occurring. Yet in Fig 1 with the recombinant alpha-synuclein and in Fig 2A with the CSF sample there is no indication that there could be a delta developing at that wavenumber. Please explain to the lay person reading this paper, what this elevation at >1700 cm⁻¹ means and why it is not considered in your analysis.

6) In Fig 4 and 7S two thresholds were selected to discriminate between low, medium and high ratio samples. Please describe either in the results or the methods how these two thresholds were established. Otherwise it appears quite random to pick two thresholds.

This reviewer would also suggest replacing Fig 4A with Fig 7S. It's more powerful and has the nice visual information of the large overlap between PD and controls in the intermediate samples of each group. The authors can use this opportunity to discuss later why there is such a high overlap and what this means for developing a diagnostic test with this method. Are the controls with intermediate or high ratio already under way to be diagnosed later?

7) Thank you for showing MSA and PD groups separately in Figure 6S. Yet, it is not conceivable to estimate and report an AUC analysis or a Mann-Whitney U test for only n = 5. Also using the term 'tendency' is in this reviewer's opinion inadequate here and it is requested to remove that sentence 'Here, a first tendency is presented' on page 12 to avoid leading the readers into a direction that is not justified with such a small n. It would be much more interesting if the authors could discuss later why the MSA cases look similar to controls in their spectra ratio although they have a clear fibrillar synucleinopathy with the respective glial inclusions, which is absent in PD. Yet the PD samples seem to have more beta-sheet content than the MSA samples.

8) The comparability results of the different instrument are mentioned in the discussion part (Fig 8S and 9S). They are worth getting prominently reported in the results section instead of in a small paragraph of the discussion. It's great to see such nice reproducibility between instruments at that stage of the instrument development.

9) Consider rephrasing the sentence 'As another example, an α Syn-PET tracer will represent an important option for PD diagnosis, similar to what they have for amyloid- β in AD.' on page 15, first paragraph of the discussion. The message in the context of that paragraph is unclear; and who is 'They'?

10) In the second paragraph of the discussion you mention 'Because misfolding is expected already in NSD stage 2 [...]'. It would be good to introduce NSD in a sentence to the lay reader as this is a very fundamentally new concept in the field of diagnosis of synucleinopathies. And why do you state 'already at NSD stage 2'? According to the NSD staging, misfolded alpha-synuclein (i.e., n-asyn) is already present at NSD stage 1 (see Dam et al., NPJ Parkinsons Dis. 2024; PMID: 39333167). Identifying this prodromal stage of PD for clinical trials or prevention therapies would be the high goal.

Referee #3 (Remarks for Author):

The authors of this manuscript introduce the immuno-infrared sensor (iRS) to detect α -synuclein misfolding in cerebrospinal fluid. The assay achieved high accuracy in distinguishing PD and MSA from controls, identifying a shift from α -helical to β -sheet conformations. This novel approach complements existing seed amplification assays by directly measuring all α -synuclein conformers.

Comments:

Could the authors please comment on the choice of producing their own α -syn antibody instead of using one of the many commercial ones? I am asking if this choice was due to assay performance or antibody rights. If it's related to antibody performance, please add these details to the text.

This reviewer cannot find the sample collection and storage details anywhere.

Apart from figure 3S, it would also be helpful to show Western blot images demonstrating the antibody's ability to bind all α -syn forms.

Although this information can be derived from parts of the text, the authors should clearly state the fluid volume required to run this assay and provide the estimated limit of detection.

Figure 2S must be improved. Is the zigzag line meant to represent the laser IR light? Please make it larger and include legends directly within the figure.

I think that apart from figure 2S and figure 2A, the manuscript would significantly benefit from including a graphical scheme of the experimental setup. I suspect that many in the neurology biomarker field are not familiar with IR Spectroscopy.

The choice of the "best measure" and thresholds seems arbitrary. For choosing that specific ratio, if you performed a grid search of all possible ratios, please explain. Otherwise, there are several algorithms that can be used to evaluate all possible wavenumbers and select the best ones (e.g., LASSO or XGBoost). For thresholds, please specify better how you determined the ones for low, intermediate, and high misfolding. Alternatively, you could use thresholds based on 95% sensitivity and specificity. In general, add more details in the statistical analysis section.

In the discussion, please consider commenting on the fact that this assay is not limited to PD and MSA but could also be applicable to AD. α -Syn co-pathology in AD has recently been shown to be an important prognostic factor (<https://doi.org/10.1002/alz.13658>, <https://www.nature.com/articles/s41467-024-52299-1>).

Letter of response to reviewer comments

Please find the individual answers to the reviewer's comments below. Changes are marked in the main text (yellow) and respective lines are provided for the corresponding Word document.

Reviewer #2:

- I. *The authors state that the alpha-synuclein antibody was provided by AC Immune. Normally, when a new antibody is published, more characterization is provided, such as the epitope, the binding affinity/KD, and binding kinetics by SPR. The authors mention on page 6 this information is available. Please provide this information. If it cannot be provided, please state clearly why not. The scientific community would profit from knowing more about this alpha-synuclein antibody and appreciates that AC Immune is making it available.*

Answer: We thank the reviewer for highlighting this important point and agree. Unfortunately, due to ongoing patenting processes we are not able to state more details about the antibody currently. It will be published elsewhere. We included this clarification in the text (p9, ll19-20).

- II. *Also, conjugating antibodies often leads to changed binding properties. For a better understanding about availability and orientation of the antibody on the chip (cartoons show an C-terminal binding of the Fc part to the chip and the Fab part looks up, is that correct?), please indicate how many DBCO molecules are labeled per antibody molecule and also show data that the DBCO-labeled antibody has the same binding properties as the unlabeled antibody.*

Answer: The reviewer is totally right, that conjugation of linker molecules to antibodies can change the binding properties of such. We tested via ELISA experiments unlabeled and labeled lots of the antibody directly in comparison. We observe comparable binding capacities for monomers and fibrils around 0.201-0.271 nM. This observation is consistent over several processes. Please see Appendix Figure S1. On the other hand, we determine the degree of labeling for the DBCO conjugation according to a published protocol (reference in *section antibody-labeling and attachment to surface*). Our threshold is around 1.5-2.5 where we accept the antibody for surface functionalization. We did extensive work on this topic to determine the right degree of labeling. However, along with comment 1 we are currently not able to share binding properties and the epitope of the antibody in detail due to ongoing patenting processes.

- III. *Also, what is the density of the antibody on the chip? In the cartoon of Fig 2S it looks densely packed whereas the cartoon in Fig 2 shows a much less densely packed surface. It is suggested using the same cartoon for illustration to avoid confusion for the read.
At a certain density one would also expect more avidity effects and possible competition*

of binding aggregated alpha-synuclein with beta-sheets over monomeric alpha-helical alpha-synuclein. How is this considered for the 1656.0/1623.5 cm⁻¹ spectral ratio calculations? Please add to the discussion.

Answer: We thank the reviewer for drawing our attention to this issue. We have revised Figure 2 so that it is in line with **Expanded View Figure EV2**. The biochip is densely packed (μg 's of antibody, molar excess to pg-ng of aSyn). The ratio is simply derived from the difference spectra on group level at points of maximal amplitude difference as qualitative separator.

III. *Figure 2A, right panel shows the effect of how alpha-synuclein was removed from the CSF sample by exposing it to the chip containing an alpha-synuclein antibody. The graph shows a prominent elevation of the absorption at 1550 cm⁻¹ and a disappearance of that spectra after removing of alpha-synuclein. Please add to the results this observation and explain what it means. Also, what happens at 1550 cm⁻¹ which was omitted on Figure 3A?*

Answer: This is a very important point. The signal at 1550 cm⁻¹ describes the amide II band which is mainly derived from the C-N stretching and the N-H bending vibration of the peptide bond. It is indicative for the relative amount of protein bound to the surface. Since we are measuring in a flow-through system the operator observes an increase of the signal in the sample circulation step because of the protein bulk flushing over the surface. This signal is always higher than the one detected in the sample wash step which is used as the result. The absorbance disappears in the red spectra because all alpha-synuclein is already removed in the first run (black spectra). In the following run the sample supernatant from the first run leads to a non-detectable signal indicating alpha-synuclein is no longer present in the sample and the first run was already efficient. For the determination of the secondary structure distribution and therefore the misfolding of alpha-synuclein the amide I band from 1600-1700 cm⁻¹ is used. For an α -helical or random coil conformer a band around 1650 cm⁻¹ is expected, whereas the band is downshifted in the oligomer (1647 cm⁻¹) and in the fibril form (1624 cm⁻¹) due to extended β -sheets. The absorbance maximum indicates the secondary structure distribution. The main readout relies on the amide I band and therefore the amide II band is not considered in Figure 3A.

IV. *Figure 3A: At >1700 cm⁻¹ an elevation of the delta signal is occurring. Yet in Fig 1 with the recombinant alpha-synuclein and in Fig 2A with the CSF sample there is no indication that there could be a delta developing at that wavenumber. Please explain to the lay person reading this paper, what this elevation at >1700 cm⁻¹ means and why it is not considered in your analysis.*

Answer: We thank the reviewer for drawing attention to this point. We did not discuss any changes >1700 cm⁻¹ before to not dilute the discussion about changes in the secondary structure. However, the changes in the region of 1700-1720 cm⁻¹ may indicate changes in unsaturated esters, ketones, or lipids that derive from metabolism changes. Since the changes are visible in the iRS, they derived from interaction partners of α Syn. The appearing band in PD/MSA individuals indicates the elevated presence of those species compared to controls. Although these are global changes without a specific assignment, we included the information

in the main text (p14, II10-13). Additionally, we have shown in another publication that lipids are part of the Lewy bodies and play a crucial role in pathology (10.1038/s41593-019-0423-2).

- V. *In Fig 4 and 7S two thresholds were selected to discriminate between low, medium and high ratio samples. Please describe either in the results or the methods how these two thresholds were established. Otherwise it appears quite random to pick two thresholds. This reviewer would also suggest replacing Fig 4A with Fig 7S. It's more powerful and has the nice visual information of the large overlap between PD and controls in the intermediate samples of each group. The authors can use this opportunity to discuss later why there is such a high overlap and what this means for developing a diagnostic test with this method. Are the controls with intermediate or high ratio already under way to be diagnosed later?*

Answer: We thank the reviewer for this comment. The two thresholds were picked in such way that AUC reached 0.95 and a minimal number of samples were assigned to the intermediate group with higher uncertainty. We have implemented your suggestion for replacing Figure 4A with 7S and added a short discussion of the overlap in the misfolding analysis (p.16, II23-29). We strive to follow up the controls with low-ratio or misfolding. However, until now, there is no follow-up analysis available.

- VI. *Thank you for showing MSA and PD groups separately in Figure 6S. Yet, it is not conceivable to estimate and report an AUC analysis or a Mann-Whitney U test for only $n = 5$. Also using the term 'tendency' is in this reviewer's opinion inadequate here and it is requested to remove that sentence 'Here, a first tendency is presented' on page 12 to avoid leading the readers into a direction that is not justified with such a small n . It would be much more interesting if the authors could discuss later why the MSA cases look similar to controls in their spectra ratio although they have a clear fibrillar synucleinopathy with the respective glial inclusions, which is absent in PD. Yet the PD samples seem to have more beta-sheet content than the MSA samples.*

Answer: We thank the reviewer for the advice and accordingly removed the text passage as well as MWU testing and AUC analysis. Instead, we show the subgroup analysis as boxplots and discussed the lower amount of b-sheet analysed in the presented MSA cases and in comparison, to the PD cases.

Literature reports of MSA cases with glial cytoplasmic inclusions (GCIs) as hallmarks have reported distinct seeding characteristics and altered b-sheet content if compared to PD pathology.⁵⁰⁻⁵² SAA data of MSA cases suggested a shorter lag phase, higher seeding efficiency, and faster aggregation rate.¹⁴ Cryo-EM structures demonstrated distinct strains for MSA aggregates before.⁴² Interestingly it was found, that the content of b-sheet structures in Lewy Bodies in patients with PD was higher than in GCIs in patients with MSA (52.6 ± 1.9 % vs. 38.1 ± 0.9 %) by Fourier-transform infrared micro-spectroscopy (FTIRM), although importantly this does not reflect quantity in CSF.⁵¹ In case of the limited presented cases ($n=5$), the ratio of the MSA subjects is closer to controls and not below PD cases (compare EV Figure 4), implying less b-sheets or other conformers in these cases. These results may align with the FTIR literature but be in contrast to other reports of the still ambiguous literature. Further direct comparisons of SAA and iRS with the same samples are strived to analyze if increased

SAA activity correlates with lower iRS ratios and vice versa or if the readout possibly relies on a different set of species and therefore is not correlating. We included these points into our discussion (p17, l26 – p18, l5).

VII. *The comparability results of the different instrument are mentioned in the discussion part (Fig 8S and 9S). They are worth getting prominently reported in the results section instead of in a small paragraph of the discussion. It's great to see such nice reproducibility between instruments at that stage of the instrument development.*

Answer: We appreciate the positive comment. Unfortunately, we cannot present as many results as we like in the main text in the form of illustrations. As this particularly important result is of a more technical nature and we wanted to focus on the application as a diagnostic tool in our main text, we decided to integrate this technical basis into the supporting information.

VIII. *Consider rephrasing the sentence 'As another example, an α Syn-PET tracer will represent an important option for PD diagnosis, similar to what they have for amyloid- β in AD.' on page 15, first paragraph of the discussion. The message in the context of that paragraph is unclear; and who is 'They'?*

Answer: We thank the reviewer for mentioning this ambiguity. The sentence was revised accordingly (p15, ll23-25).

IX. *In the second paragraph of the discussion you mention 'Because misfolding is expected already in NSD stage 2 [...]'. It would be good to introduce NSD in a sentence to the lay reader as this is a very fundamentally new concept in the field of diagnosis of synucleinopathies. And why do you state 'already at NSD stage 2'? According to the NSD staging, misfolded alpha-synuclein (i.e., n- α syn) is already present at NSD stage 1 (see Dam et al., NPJ Parkinsons Dis. 2024; PMID: 39333167). Identifying this prodromal stage of PD for clinical trials or prevention therapies would be the high goal.*

Answer: We thank the reviewer for this comment. We try to clarify this point by adding a brief description of the NSD staging system in the introduction. (p2, ll8-14) We exchanged our false statement about aSyn misfolding in NSD stage 2 to NSD stage 1, since, as correctly stated by the reviewer, it may occur in stage 1 and be utilized as a prodromal marker in the future. (p15, ll31-33)

Reviewer #3:

- I. *Could the authors please comment on the choice of producing their own α -syn antibody instead of using one of the many commercial ones? I am asking if this choice was due to assay performance or antibody rights. If it's related to antibody performance, please add these details to the text.*

Answer: We thank the reviewer for this question. We tested different antibodies including commercially available ones in our setup, which requires labeling of the antibody, and found the candidate favorable. Unfortunately, due to ongoing patenting processes we cannot state more details about the antibody performance currently. It will be published elsewhere. We included this clarification in the text (p9, ll19-20).

- II. *This reviewer cannot find the sample collection and storage details anywhere.*

Answer: Thank you for the comment. We added the storage conditions at the site of our iRS analysis to the description, which can be found in the study cohorts section of the methods part (p5, ll28-32). Samples were kept at -80 °C in ready-to-use aliquots, transferred under dry ice conditions from clinics to the side of the iRS-measurement, and stored there at -80 °C until further use.

- III. *Apart from figure 3S, it would also be helpful to show Western blot images demonstrating the antibody's ability to bind all α -syn forms.*

Answer: We thank the reviewer for this suggestion. We performed native PAGEs and Western Blot studies with self-made time-dependent antigens extracted from an aSyn fibrillation series (0-144, intermediates taken each 24h) and observed the formation of different species covering the whole range of the native gel and binding to the study antibody. Since we showed ELISA and ThT data for three commercially available antigens, we additionally included Dot blotting results on those three forms. For the three commercial species did not perform Western Blots, which additionally show the molecular weights of the species, since these were available from the manufacturer and the ThT data resolved the presence of b-sheets and aggregates in the case of the pre-formed fibrils (compare EV Figure 1). As our infrared and ThT analyses already implicate the binding of different antigens, we would refrain from implementing it either in the main or supplement texts. However, for a complete answer of this reviewer point, we show the results for Dot blot and Western blot right below.

(A) Dot-Blot with anti- α Syn antibody

(B) Western Blot with anti- α Syn antibody of native gel

Figure LoR1: Panel A: Dot Blot analysis of the reactivity and cross-reactivity of the study antibody Ab-towards α Syn and Ab-antigens and human serum albumin (HSA). The antigens were applied to a nitrocellulose membrane, blocked, and incubated with the study antibody (1:1000 dilution), followed by incubation of secondary antibodies (anti-mouse, 1:5000 dilution), which were enzyme-coupled for substrate conversion and color reaction. 2 μ g antigen was applied/spot. Reactions against 8 samples were tested, as listed in the legend. Panel B: Native PAGE Western Blots of α Syn fibrillation intermediates (t=0-144 h) under physiological conditions (137 mM NaCl, 7.8 mM Na₂HPO₄, 1.4 mM NaH₂PO₄, 3.2 mM KCl, pH 7.4). During fibrillation, especially visible at t=72h and later (t=144 h), larger aggregates are formed with sizes >> 100 kDa ranging over the whole range of the used gel (15-10000 kDa, Invitrogen™ NativePAGE™ Bis-Tris Gel, art. no. 10799853). The exact sizes cannot be stated due to the lack of a feasible high-molecular weight marker. NativePAGE and Western Blot were performed according to manufacturer's protocol. 10 μ g of protein was loaded in each column. Prestained marker: PageRuler™ (ThermoFisher Scientific, cat. no. 26617).

IV. *Although this information can be derived from parts of the text, the authors should clearly state the fluid volume required to run this assay and provide the estimated limit of detection.*

Answer: We thank the reviewer for this constructive suggestion. We calculated the LoD to 157 μ l CSF. We revised Figure 5S (Panel B) and stated the CSF volume (300 μ l) used for each sample analysis. (EV3, p26, II10-12)

V. *Figure 2S must be improved. Is the zigzag line meant to represent the laser IR light? Please make it larger and include legends directly within the figure. I think that apart from figure 2S and figure 2A, the manuscript would significantly benefit from including a graphical scheme of the experimental setup. I suspect that many in the neurology biomarker field are not familiar with IR Spectroscopy.*

Answer: We thank the reviewer for requesting clarifications regarding the setup. We updated EV Figure 2 to include a scheme of the setup, including the flow internal reflection element, the flow chamber, and the evanescent field protruding in the medium with the functionalized surface at the sides of the multiple total reflections of the IR light. The evanescent field marks the measurement area with a penetration depth of roughly 500 nm. We also included a surface functionalization layer scheme (Panel B). Further details of the approach and measurement process are described in multiple publications, for example in Nabers et al. (10.1002/jbio.201400145), where the setup including tests for the density of the layers are described.

- VI. *The choice of the "best measure" and thresholds seems arbitrary. For choosing that specific ratio, if you performed a grid search of all possible ratios, please explain. Otherwise, there are several algorithms that can be used to evaluate all possible wavenumbers and select the best ones (e.g., LASSO or XGBoost). For thresholds, please specify better how you determined the ones for low, intermediate, and high misfolding. Alternatively, you could use thresholds based on 95% sensitivity and specificity. In general, add more details in the statistical analysis section.*

Answer: The best measure was assessed by calculation of the group difference spectra and subsequently picking the ratio showing the largest differences between both groups (1656.0/1623.5-R, compare Figure 3). We tested other spectral ratios including 1650/1630-R, 1651.5/1630-R, 1651.5/1623.5-R, but found their ROC-AUC performance worse in comparison to the 1656.0/1623.5-R, which is congruent with the difference spectrum since the difference at the alternative wavenumber ratios is not maximal. We extensively tested linear regression models of distinct ratios and other measures (e.g. absolute amide-I maximum) However, they did not yield an enhanced performance, compared to the 1656.0/1623.5-R alone. Moreover, machine learning approaches (Random Forest, support vector machines, etc.) were performed to train classifiers for PD/MSA vs. DC separation, but these were not superior to the data-derived single ratio of 1656.0/1623.5. We excluded machine learning approaches based on deep learning because the dataset was too small. That is why we selected this 1656.0/1623.5-R as "best measure". The two thresholds were picked in such way that AUC reached 0.95 and a minimal number of samples were assigned to the intermediate group with higher uncertainty.

- VII. *In the discussion, please consider commenting on the fact that this assay is not limited to PD and MSA but could also be applicable to AD. α -Syn co-pathology in AD has recently been shown to be an important prognostic factor (<https://doi.org/10.1002/alz.13658>, <https://www.nature.com/articles/s41467-024-52299-1>).*

Answer: We thank the reviewer for this suggestion. We included the reference in our discussion.

4th Mar 2025

Dear Klaus,

Thank you for submitting your revised manuscript to EMBO Molecular Medicine. We have now received the enclosed report from the referee who agreed to re-assess it. As you will see, the referee is overall satisfied with the revisions and I am pleased to inform you that we will be able to accept your manuscript pending the following amendments.

1. We understand that the antibody used in the study is currently undergoing the patent approval process, and we also acknowledge that the reviewer was satisfied with your response on this matter. However, in accordance with our author guidelines and our goal of publishing studies that are both interpretable and reproducible, we kindly request that you provide the epitope sequence of the new antibody used in the study in the Reagents and Tools table.
2. In the Data Availability section, please add the following statement: "The alpha-synuclein antibody used in the research is available to colleagues in academic research upon request to AC Immune."
3. Please remove the figures from the manuscript file. The figure legends should remain in the manuscript.
4. Table EV1, along with its legend, should be removed from the manuscript file and uploaded separately as an xls file using the "Expanded View" file type. Since the table is rather complex, please update it to Dataset EV1. The legend can be included in the same file on a separate sheet labeled "Legend." Callouts need to be updated accordingly.
5. In Methods (and in the author checklist), for human studies, please include a statement the experiments conformed to the principles set out in the WMA Declaration of Helsinki and the Department of Health and Human Services Belmont Report.
6. Please provide up to five keywords.
7. "The paper explained" should be moved to the main manuscript file and divided into three sections, with sections headings: "problem", "results", "impact".
8. I have made some modifications to the synopsis text(see attached). Please let me know if it's fine as is.
9. Please resolve the author name discrepancy - 'Mollenhauer' in the manuscript versus 'Brit Moellenhauer' in the submission system. Additionally, the corresponding author needs to be clearly labeled on the manuscript title page, along with his email address provided on the title page.
10. Funding information needs to be part of Acknowledgements section.
11. The references need to be formatted according to the EMBO Molecular Medicine reference style. Please list up to 10 co-authors of a paper before adding et al. in the reference list. Citations should be listed in alphabetical order. "Main Text" should be removed from the section heading.
12. "Competing interests" should be renamed to " Disclosure and Competing Interests Statement".
13. Please resolve the following issues regarding callouts:
 - callout for Figure 1A is missing.
 - "EV Figure 2", "EV Figure 4" should be renamed to "Figure EV2", "Figure EV4".
 - "EV Table 1S" needs to be 'Dataset EV1'.
 - a callout for Appendix Figure S4 is missing.
14. Appendix: the separately uploaded figures and the Word version should be removed as we only need the Appendix file (with all the Appendix items) to be in one single PDF file; page numbers are also needed throughout the file and in the Table of Content on the title page.
15. At EMBO Press we require authors to provide source data for the main manuscript figures. Our source data coordinator reached out to you on December 13th, 2024 to discuss which figure panels we would need source data for and provided helpful tips on how to upload and organize the files. Please submit the requested source data with your next submission.
16. Please provide a visual abstract to illustrate your article as a PNG file 550 px wide x 300-600 px high. Please kindly ensure the file size does not exceed these requirements.

18. The manuscript sections should be in the following order: Title page - Abstract & Keywords - Introduction - Results - Discussion - Methods - Data Availability - Acknowledgments - Disclosure Statement & Competing Interests - References - Figure Legends -(Main Tables with legends if applicable) - Expanded View Figure Legends

19. The nomenclature of EV figures in the legend titles (manuscript file) is not correct: needs to be Figure EV1, etc. instead of EV Figure 1, etc.

20. Materials & Methods should be 'Methods'.

21. Please address the following issues related to figure legends

- Please note that the exact p values are not provided in the legends of figures 3B, EV5, supplementary figure 2.
- Please indicate the statistical test used for data analysis in the legends of supplementary figure 2.
- Please note that the box plots need to be defined in terms of minima, maxima in the legend of figure 3B, 4A, EV4
- Please note that the box plots need to be defined in terms of minima, maxima, centre, bounds of box and whiskers, and percentile in the legend of figure EV5.
- Please note that information related to n is missing in the legend of figure 2B, EV1 B
- Please note that the error bars are not defined in the legend of figure EV1 B.
- Please note that the measure of center for the error bars needs to be defined in the legend of figure 2B.

I look forward to seeing a revised form of your manuscript as soon as possible.

Kind regards,
Jingyi

Jingyi Hou
Senior Editor
EMBO Molecular Medicine

*** Instructions to submit your revised manuscript ***

***** Reviewer's comments *****

Referee #2 (Comments on Novelty/Model System for Author):

The applied mathematical modelling appears adequate at this early stage of the nonclinical development of this new biosensor (i.e., Proof-of-Concept). After measuring more samples and in a diagnostic GMP/GLP setting the model will need to be adapted to better suit clinical application.

Referee #2 (Remarks for Author):

Thank you for your clear responses to the questions and considering the proposed changes.

This reviewer understand the difficulties with publishing the antibody characteristics when a patent application is in process in parallel. Hopefully, this team and AC Immune can soon publish the missing information about the used antibody in order for the field to be able to expand the conclusions from this study and biosensor development.

The authors addressed the remaining editorial issues.

27th Mar 2025

Dear Klaus,

Thank you for sending us your revised manuscript. We are pleased to inform you that your manuscript is accepted for publication and is now being sent to our publisher to be included in the next available issue of EMBO Molecular Medicine.

Yours sincerely,
Jingyi

Jingyi Hou
Senior Editor
EMBO Molecular Medicine
